# The EUREC[4]A turbulence dataset derived from the SAFIRE ATR 42 aircraft

Pierre-Etienne Brilouet[1], Marie Lothon[1], Jean-Claude Etienne[2], Pascal Richard[2], Sandrine Bony[4], Julien Lernoult[3], Hubert Bellec[3], Gilles Vergez[3], Thierry Perrin[3], Julien Delanoë[5], Tetyana Jiang[3], Frédéric Pouvesle[3], Claude Lainard[3], Michel Cluzeau[3], Laurent Guiraud[3], Patrice Medina[1], and Theotime Charoy[3]

[1]Laboratoire d'Aérologie, University of Toulouse, CNRS, UPS, Toulouse
[2]CNRM, Météo-France, CNRS, Toulouse
[3]SAFIRE, CNRS, Météo-France, CNES, Toulouse
[4]LMD/IPSL, CNRS, Sorbonne University, Paris, France
[5]Laboratoire Atmosphères, Milieux, Observations Spatiales/UVSQ/CNRS/UPMC, Guyancourt, France

**Correspondence:** Marie Lothon (marie.lothon@aero.obs-mip.fr)

**Abstract.**

During the EUREC[4]A field experiment that took place over the tropical Atlantic ocean, East of Barbados, the French ATR 42 environment research aircraft of SAFIRE aimed at characterizing the shallow cloud properties near cloud-base, and the turbulent structure of the subcloud-layer. For this purpose, the aircraft payload included radar and lidar remote sensing,
microphysical probes, a laser spectrometer, and meteorological sensors. In particular, the aicraft was equipped with a 5-hole radome nose and several temperature and moisture sensors allowing for measurements of wind, temperature and humidity at 25 Hz. This paper presents the high frequency measurements made with these sensors and their translation in terms of turbulent fluctuations, turbulent moments and characteristic length scales of the turbulence. A particular focus is put on the calibration and the quality control of the air moisture measurements, which remain a challenge at fine scales. Level-2 and Level-3 data
are distributed as an ensemble of NetCDF files available to the public at AERIS (https://doi.org/10.25326/128 ; Brilouet et al., 2020).

*Copyright statement.* TEXT

## 1 Introduction

For many decades, difficulties in quantifying the strength of the low-level cloud feedback, especially in the trade-wind regions,
have hindered precise estimates of the climate sensitivity. Improving our estimate of the feedback requires to better understand the physical processes that control cloudiness in the trades, and their dependence on environmental conditions. The low-level clouds that form in the trade-wind regimes are closely associated with shallow cumulus convection, and early studies by Malkus (1977) and LeMone and Pennell (1976) showed that the properties of trade-cumuli could be understood to a large extent by

examining the properties of the subcloud layer. Moreover, trade-wind clouds are known to organize in various mesoscale patterns (referred to as 'Sugar', 'Gravel', 'Fish' or Flowers') that embed different cloud types and depend on environmental conditions (Stevens et al., 2020; Bony et al., 2020). LeMone and Pennell (1976) and LeMone and Meitin (1984) suggested that the shallow cloud organization could also be rooted in the structure of the subcloud layer. For instance, LeMone and Pennell (1976) showed that in highly suppressed conditions, the cloud distribution was related to the organization of the subcloud layer in structures such as roll vortices. In constrast, in situations of enhanced convection the turbulence seemed to be more directly and locally linked to individual clouds. The state of the subcloud layer thus seems to influence the degree of coupling (or decoupling) between the surface and clouds, and thus the cloud distribution. However, accurate and intensive observations are needed to further elucidate the connection between the mesoscale cloud patterns, the subcloud layer and the surface.

The EUREC[4]A field campaign was designed to better understand what controls the trade-wind cloudiness, its mesoscale organization, and its interplay with convection and circulations over a wide range of scales (Bony et al., 2017). The experiment took place in January-February 2020 over the tropical western Atlantic, East of Barbados. Many observing systems were deployed during the campaign, including four research aircraft, four research vessels, and a large number of autonomous observing systems in the ocean and in the atmosphere (Stevens et al., 2021). One of the aircraft was the ATR 42 operated by the French Research Aircraft Infrastructure for Environmental Studies (SAFIRE). During the campaign, its mission was primarily devoted to the characterization of the shallow cloudiness near cloud-base (Chazette et al., 2020; Delanoe, 2021) and the turbulent properties of the subcloud layer. Its flights were closely coordinated with those from HALO, the German research aircraft, which was flying large circles at a higher altitude to observe clouds from above and to characterize the dynamical and thermodynamical environment through intensive dropsonde measurements (Konow, 2021). The characterization of the turbulence within the Marine Atmospheric Boundary Layer (MABL) plays a major role in EUREC[4]A, as it will help decipher the interactions between turbulence, convection and clouds, as well as the dependence of clouds on surface and large-scale conditions. Moreover, the MABL being the interface between the ocean surface and the cloud layer, the characterization of its turbulent structures should also help understand how mesoscale and sub-mesoscale heterogeneities at the ocean surface, associated with the presence of ocean eddies or sea surface temperature fronts, could imprint themselves in the cloud organization aloft.

This paper describes the EUREC[4]A dataset containing the turbulent fluctuations and turbulent moments associated with the high frequency measurements of temperature, moisture and wind from the SAFIRE ATR 42 aircraft, computed over horizontal stabilized legs. Section 2 describes the flight strategy and the type of meteorological and cloud conditions encountered during the flights. Section 3 presents the in-situ instrumentation. Sections 4 and 5 explain the quality control procedure and the calibration methodology used to process the moisture and temperature fluctuations. Section 6 explains how the turbulent moments are computed, and how their systematic and random errors are quantified. Length scales characteristic of the turbulent field are also estimated. Section 7 describes the turbulence dataset in more details, and shows a few illustrations of its content. A conclusion is given in Section 8.

## 2 Flight strategy and conditions

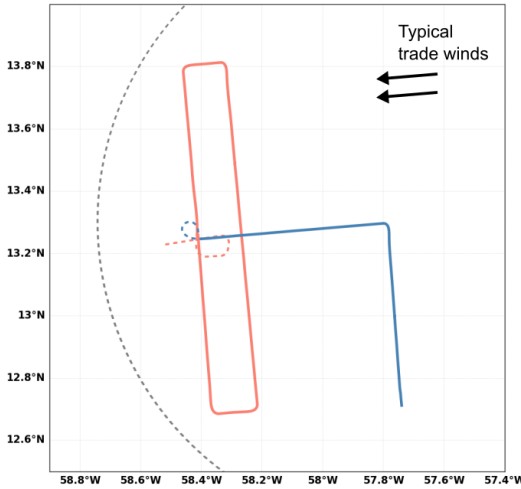

**Figure 1.** Schematic horizontal trajectory of the SAFIRE ATR 42 within the HALO circle (in gray dashed-lines) during EUREC[4]A. Maneuvers for alignment, ferry legs and other parts of the trajectory projection are not shown here. R-pattern is shown in red and L-pattern is shown in blue.

The core experimental strategy of the EUREC[4]A field campaign was based on the coordination of the SAFIRE ATR 42 and HALO aircraft: while HALO was flying large circles (200 km diameter, referred to as EUREC[4]A circles) at an altitude of about 9 km (Konow, 2021), the SAFIRE ATR 42 was flying in the lower troposphere of the western half of the circle, describing two types of pattern (Fig. 1):

- a 'R-pattern composed of at least two rectangles (of about 120 km by 15 km) flown at cloud base, to characterize the cloud-base cloud fraction through horizontally-staring lidar and radar measurements

- a 'L pattern' flown within the subcloud layer at two different heights, to characterize the turbulence and the coherent structures of the boundary layer.

At the end of most flights, these two patterns were completed by a short surface leg ('S leg') flown at 60 m above sea level, before returning to the airport.

During each SAFIRE ATR 42 flight, 2 to 4 rectangles were flown, generally around the cloud-base level, except when stratiform clouds were occurring higher up and a rectangle was also flown around the trade-inversion level. In addition, 2 to 4 L-patterns were flown, each pattern being composed of 2 straight legs of about 60 km (one along wind and one across wind) flown either near the top or the middle of the subcloud layer. These patterns aimed at exploring the anisotropy of the turbulence and the organization, as well as the vertical structure of the boundary layer.

During a single HALO flight of about 9 hours, the ATR 42 flew 2 flights (using a similar flight plan) with a short refueling in between. The repetitiveness of the flight plans makes it possible to consider all the flights as being members of the same statistical ensemble.

Table 1 describes each flight plan, together with some information about the mean wind within the subcloud layer, and the types of clouds observed. It shows how, during the campaign, the conditions evolved from suppressed conditions, with only rare and thin cumulus clouds, toward more cloudiness and more vertical development. This was associated with a gradual strengthening of the mean wind in the subcloud layer. Note that the flights RF01 and RF02 are not included in the table because they were Electromagnetic Interference (EMI) and test flights, and RF20 had no rapid measurements because of an INS failure.

**Table 1.** Flight plan and wind conditions associated with each flight during the EUREC4A field campaign. The flight altitude is indicated between brackets and the "cb", "strati" and "surf" notations refer to "cloud base", "stratiform layer" and "surface", respectively. The wind conditions within the subcloud layer are inferred from the averaged airborne measurements over the L-legs, including both top- and mid-subcloud layer (except for RF16, during which there was no L-pattern). $t_i$ and $t_f$ are UTC times of start and end of the flights respectively.

| RF | Date | $t_i$ | $t_f$ | Flight strategy | Wind conditions | Cloud cover |
|----|------|-------|-------|-----------------|-----------------|-------------|
| 03 | 26/01 | 11 59 | 16 04 | 2R$_{cb}$ (800 m) <br> 3L (580 m, 400 m, 60 m) | $7.2 \pm 0.8$ m.s$^{-1}$ | Small cloudiness <br> Scarce and very thin Cu |
| 04 | 26/01 | 16 57 | 21 26 | 3R$_{cb}$ (800 m) <br> 2L (600 m, 400 m) + L$_{surf}$ (60 m) | $3.5 \pm 0.7$ m.s$^{-1}$ | Small cloudiness <br> Scarce and very thin Cu |
| 05 | 28/01 | 20 36 | 24 50 | 2R$_{cb}$ (670 m) <br> 2L (450 m, 300 m) + L$_{strati}$ (1800 m) | $7.8 \pm 0.6$ m.s$^{-1}$ | Small cloudiness <br> Getting smaller with time |
| 06 | 30/01 | 11 11 | 15 31 | 3R$_{cb}$ (650 m - 750 m) <br> 2L (550 m, 300 m) + L$_{surf}$ (60 m) | $8.9 \pm 0.5$ m.s$^{-1}$ | Scattered ShCu <br> Less cloudy with time |
| 07 | 31/01 | 14 59 | 18 48 | 2R$_{cb}$ (600 m) + 1/2R$_{cb}$ (700 m) <br> 2L (400 m, 200 m) | $7.9 \pm 0.7$ m.s$^{-1}$ | Sparse Cu at the <br> edge of cold pools |
| 08 | 31/01 | 19 49 | 24 01 | 3R$_{cb}$ (550 m - 610 m) <br> 2L (450 m, 300 m) | $6.6 \pm 0.6$ m.s$^{-1}$ | Mostly ShCu <br> Few towering clouds |
| 09 | 02/02 | 11 34 | 15 37 | 2R$_{cb}$ (600 m) <br> 2L (300 m, 600 m) + 1/2L (600m) + L$_{strati}$ (1100 m) | $7.4 \pm 1.4$ m.s$^{-1}$ | Heterogeneous |
| 10 | 02/02 | 16 44 | 21 03 | 3R$_{cb}$ (710 m - 770 m) <br> 2L (580 m, 300 m) + L$_{surf}$ (60 m) | $5.4 \pm 0.8$ m.s$^{-1}$ | Heterogeneous |
| 11 | 05/02 | 08 45 | 12 59 | R$_{strati}$ (1830m) + 2R$_{cb}$ (680 m - 740 m) <br> 2L (530 m, 250 m) + L$_{surf}$ (60 m) | $10.8 \pm 1.0$ m.s$^{-1}$ | Succession of <br> Flower clouds |
| 12 | 05/02 | 13 48 | 18 04 | 3R$_{cb}$ (790 m) <br> 2L (550 m, 250 m) + L$_{surf}$ (60 m) | $9.4 \pm 0.4$ m.s$^{-1}$ | Increase of the <br> cloud fraction |
| 13 | 07/02 | 11 30 | 15 51 | R$_{strati}$ (2100 m) + 2R$_{cb}$ (750 m) <br> 2L (540 m, 60 m) + 2L$_{1/2}$ (280 m, 575 m) | $12.3 \pm 1.0$ m.s$^{-1}$ | Heterogeneous |
| 14 | 07/02 | 17 20 | 21 42 | 3R$_{cb}$ (980 m - 775 m) <br> 2L (640 m, 330 m) + L$_{surf}$ (60 m) | $11.0 \pm 0.4$ m.s$^{-1}$ | Heterogeneous |
| 15 | 09/02 | 08 37 | 13 08 | 3R$_{cb}$ (820 m) <br> 2L (620 m, 300 m) + L$_{surf}$ (60 m) | $10.9 \pm 0.4$ m.s$^{-1}$ | Patchy clouds <br> Few cold pools and strati layers |
| 16 | 09/02 | 14 03 | 18 23 | 4R$_{cb}$ (800 m) <br> L$_{surf}$ (60 m) | $11.8 \pm 0.7$ m.s$^{-1}$ | Patchy clouds <br> Few cold pools and strati layers |
| 17 | 11/02 | 05 55 | 10 21 | R$_{strati}$ (1800 m) + 2R$_{cb}$ (700 m) <br> 2L (570 m, 285 m) | $12.5 \pm 0.7$ m.s$^{-1}$ | Active convective cells |
| 18 | 11/02 | 11 30 | 15 50 | 3R$_{cb}$ (775 m) <br> 2L (560 m, 265 m) + L$_{surf}$ (60 m) | $10.9 \pm 0.4$ m.s$^{-1}$ | Multi-layers ShCu, <br> towers Cu and Strati layers |
| 19 | 13/02 | 07 35 | 11 51 | R$_{strati}$ (1850 m) + 2R$_{cb}$ (790 m - 855 m) <br> 2L (600 m, 300 m) | $12.8 \pm 0.7$ m.s$^{-1}$ | Cloud conditions with <br> StCu and towering Cu |

For the description of the cloud cover, the abbreviations are defined as follow Cu : Cumulus, ShCu : Shallow Cumulus, StCu : Stratocumulus and Flower clouds : Circular clumped patterns as introduced by Stevens et al. (2020).

## 3 Aircraft in situ instrumentation for high rate thermodynamical measurements

The SAFIRE ATR 42 is a turboprop airplane initially used for commercial aviation, which has been profoundly modified for the purpose of atmospheric and environmental research. It is permanently instrumented with in situ basic measurements (thermodynamics, radiation, microphysics), and has in addition a large flexible payload capacity which enables the use of a large number of in situ and remote sensing observations.

The core in situ instrumentation used during EUREC[4]A, and the low-rate measurements (1Hz) associated with it, are described in Etienne et al. (2021). The higher rate measurements (25 Hz) are, to a large extent, based on the same instrumentation.

The SAFIRE ATR 42 is equipped for high rate measurements of the three wind components of the air motion, air temperature and air moisture. Initially acquired at various higher sampling rates consistent with the time response of the sensors, the final high rate measurements of the meteorological variables were sampled at a common frequency of 25Hz. For a true airspeed of about $100\,\mathrm{ms}^{-1}$, this corresponds to a sample spacing of approximately 4 m.

The three components of the wind are obtained by adding the velocity vector of the aircraft with respect to the Earth, and the velocity vector of the air with respect to the aircraft. The ground velocity is measured with inertial navigation unit (AIRINS, model 6005214 from Ixblue company). The velocity of the air relative to the aircraft is computed from the measurement of the true air speed magnitude, the attack and side slip angles, according to Lenschow (1986). The attack and side slip angles are respectively deduced from the vertically aligned and horizontally aligned differential pressure measured on the five-hole nose radome with pressure transducers, according to the technique first described by Brown et al. (1983). The true airspeed ($TAS$) is calculated from the measurement of the dynamical pressure and the static pressure. The static pressure is measured on the fuselage side with a Pitot tube and a pressure transducer. The dynamical pressure is obtained by substracting the static pressure to the total pressure measured at the central radome hole. The velocity measurement and computation has proved reliable in numerous field campaigns (Lambert and Durand, 1998; Saïd et al., 2005, 2010).

Air temperature is retrieved from a platinum wire thermometer placed in a Rosemount housing (E102AL Rosemount), after correction for the adiabatic heating due to the airspeed of the plane. During EUREC[4]A, temperature was also measured using two fine wires (Baehr et al., 2002) that were housed in a tubular antenna. The two platinum fine wires are housed in a tubular antenna from SFIM company (model T4113). They are more directly exposed to the stream, but protected from radiation, which consequently should not have a significant impact.

Moisture fluctuations were measured with a Krypton Hygrometer Campbell KH20 which has been adapted for airplanes. Initially used for measurements on ground towers, this sensor was profoundly modified to be inserted into the housing of a former moisture sensor (Lyman-alpha hygrometer). The signal is calibrated based on reference slow (1 Hz) measurements of humidity. Here we use the Water Vapour Sensing System (WVSS2) for reference (Etienne et al., 2021), instead of the typical chilled-mirror sensor reference (General Eastern 1011). A Li7500 Licor sensor was used as spare for fast humidity measurements. It was also adapted for aircraft measurements and previously used in HyMeX (Estournel et al., 2016) and DACCIWA (Knippertz et al., 2015) field campaigns.

Due to several circumstances, some technical difficulties were encountered during the field campaign, especially during its first phase. In particular, a major issue concerned one of the radome pressure transducers, making it impossible to calculate the attack angle with the usual methodology. This strongly impacts the air vertical velocity estimates. As a consequence, and due to the sensitivity of air motion measurements, the dataset discussed here does not include the vertical velocity for flights RF02 to RF08, nor any estimate related to it.

The KH20 also showed issues during this first phase, partly due to the particular conditions of the marine environment encountered during EUREC[4]A, which make it challenging to measure air moisture at fine scale. The drastic change of water vapour content from above the inversion (where relative humidity can be as dry as a few percent) to below cloud base (where relative humidity is generally higher than 80%), was a challenge and the spacing between the emitter and the receiver of the KH20 sensor has been adjusted. In the subcloud layer patterns, the sea salt loading of the KH20 sensor generated a significant loss of signal dynamics. An assiduous cleaning of the optics at the beginning of each flight allowed to limit this loss of signal. Regarding the KH20 behaviour, many technical issues have been gradually solved and several improvements have been made following the feedbacks at the end of each flight. Thus, the KH20 performances have been significantly improved by the second phase of the campaign (flights RF09 to RF19). The calibration of moisture fluctuations, choice of reference slow measurement and the relative performances of the KH20 and Licor are discussed further in Section 4.

As a consequence of those difficulties, and after quality control, there are flagged or rejected data within the dataset. The second phase of the field, corresponding to flights RF09 to RF19, had much better quality data.

## 4 Calibration and qualification of the fast humidity sensors

One of the current challenges of atmospheric turbulence measurements is the fast measurement of humidity, which remains difficult at frequencies higher than 1 Hz. For many years in the past, a robust and high performance Krypton hygrometer called "Lyman-alpha" was commonly used for the measurement of (uncalibrated) air moisture fluctuations (Buck, 1976; Saïd et al., 2010; Canut et al., 2010). Since the UV source of this sensor is not available anymore, one had to use another sensor. However, reaching a similar performance remains a challenge. Here, we use a KH20 Krypton hygrometer, which has been recently adapted and installed onboard the SAFIRE ATR 42.

In this section, we discuss the data calibration and control over stabilized legs of 5 min. This segmentation is a compromise to ensure best sampling representativity and homogeneity (see Section 6 for more details).

The time series shown on Fig. 2a illustrates a comparison of uncalibrated fast measurements from the KH20 sensor with two slow sensors: the WVSS2 sensor and the 1011C miror hygrometer. The WVSS2 measures the relative humidity in % by absorption spectroscopy with a tunable diode. The 1011C hygrometer is a condensation hygrometer which measures the dew point temperature. Absolute, relative and specific humidity are then inferred from dew point, temperature and pressure. To calibrate the fast sensor with the reference slow measurement of absolute humidity, both the slow and fast signals are initially low-pass filtered at 1/6 Hz, and then a linear regression is computed to obtain the calibration slope and the intercept to be applied to the fast signal. The quality of the calibration is assessed by the R-square ($R^2$) of the linear regression between the

low-pass signals of the reference sensor and of the fast sensor. One expects $R^2$ larger than 0.98 for high quality signals of slow and fast measurements. Figure 2b shows the resulting calibrated signal, converted to water vapour miwing ratio, and compared to the slow series of the same variable.

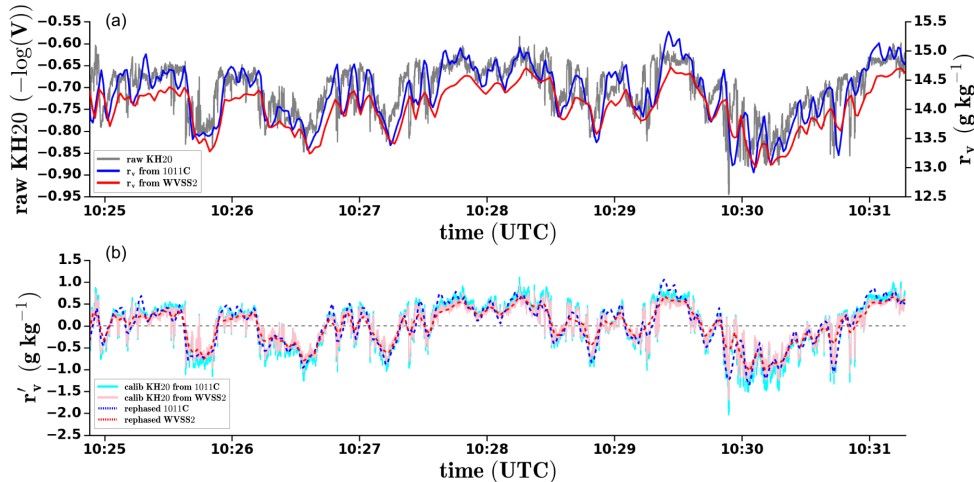

**Figure 2.** Example of humidity measurements and calibration during a leg (l1a) flown at z∼600 m on 13 February 2020 (RF19) : (a) time series of (gray line) the raw uncalibrated signal of the fast KH20 sensor, and the water vapour mixing ratio inferred from (blue line) the 1011C mirror hygrometer, or (red line) the WVSS2 sensor; (b) corresponding time series of the water vapour mixing ratio fast fluctuations derived from the KH20 calibrated signal, and the reference slow measurements, phased in time with the fast sensor, from (cyan line) the 1011C mirror hygrometer and (pink line) the WVSS2 sensor.

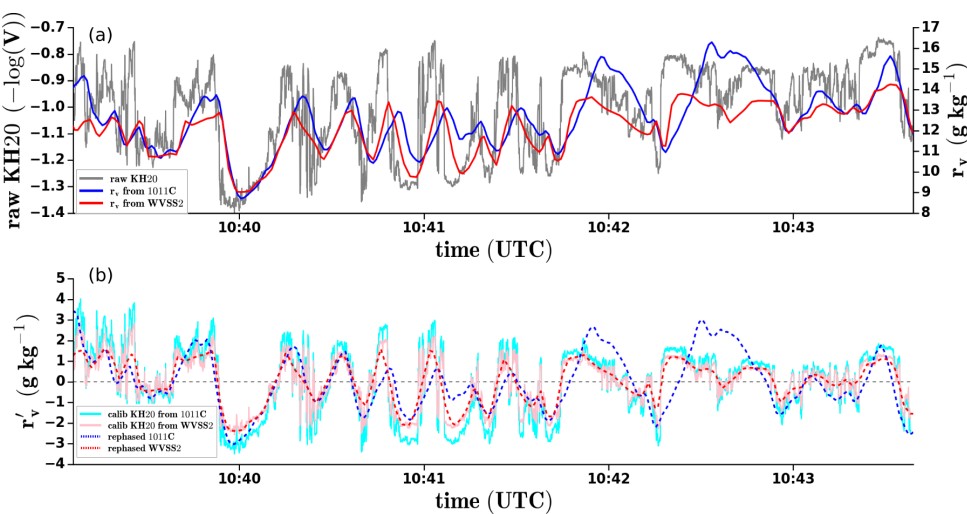

**Figure 3.** Same as Fig. 2 for a leg (l1c) flown at z∼600 m on 13 February 2020 (RF19).

## 4.1 Choice of slow sensor

The measurements of the two slow sensors exhibit differences which can impact the calibration. To optimize the choice of the slow reference and the calibration process, we considered the second phase of the campaign (flights RF09 to RF19) which had less technical issues and during which the KH20 showed a very good behaviour, in terms of time response, and of consistency with other moisture sensors. Figure 4a and 4b show the distributions of $R^2$ on all segments of flights RF09 to RF19, when using either the 1011C miror hygrometer or the WVSS2 sensor as a reference, respectively. The $R^2$ values are significantly higher when the WVSS2 sensor is used as a reference. This reveals a better behaviour of the WVSS2 sensor relatively to the 1011C hygrometer. The latter, despite its smaller response time, showed more difficulties in following the large variability of air moisture encountered during EUREC[4]A, which added to the challenges of measuring air moisture in an environment with sea salt, clouds or even rain. This phenomenon can be noticed around 10:29:20 UTC in Fig. 2b and more clearly in Fig. 3, where the 1011C signal (dashed blue) shows several exaggerated peaks, because it responded too slowly to the increasing and following fast levelling of moisture. This behaviour is explained by its measurement principle, with condensation at the mirror surface, which requires time to recover by drying. This issue resulted in a positive bias of about 27 % in the estimated moisture variance when the KH20 was calibrated with the 1011C hygrometer. This bias is visible in Fig. 2b and even more clearly in Fig. 3b from the difference of fluctuation energy between the two signals.

Figure 4 makes the distinction between legs flown within the subcloud layer (or MABL legs, associated with more homogeneous turbulence) and the rest of the legs. At cloud base, the turbulence is highly heterogeneous, with a mix of cloudy air, subcloud layer air and free tropospheric air. On the other hand, the legs flown higher up near the trade-inversion level exhibit a very weak turbulence, or an intermittent turbulence associated with individual clouds. The distributions do not show strong differences between one set and the other. This indicates that the calibration against WVSS2 actually works both in the MABL and above the subcloud layer.

The WVSS2 is a slower sensor than the 1011C hygrometer, with a time response of about 2.5 s, against about 1 s. Due to this significant delay (that can be seen in Fig. 2a), we tested the impact of phasing the slow signal to the fast signal. Figure 4c shows the significant improvement obtained with this phasing: for most of the legs, $R^2$ is now larger than 0.95. Figure 2b shows the calibrated signal of the KH20 converted in water vapour mixing ratio, along with the phased slow signal used for optimum calibration. Figure 5 shows that the phasing has only a small impact on the variance of moisture: it is only 1.7 % larger in the case of phased slow signal, which is much smaller than the random error.

For a thorough qualification of the fast moisture measurements during EUREC[4]A, considering R-square values is not sufficient. Indeed, even if the correlation with the slow signal is good, the sensor might not show the proper dynamics of amplitude of the fluctuations (e.g. due to sea salt or to an inappropriate spacing between the emitter and the receiver). For this reason, we used as an additional index, the Root Mean Squared Error (RMSE), calculated between the low-pass filtered calibrated signal (1/6 Hz) and the slow reference signal (also low-passed). The smaller this index, the better the agreement between the fast and slow sensors at large scales.

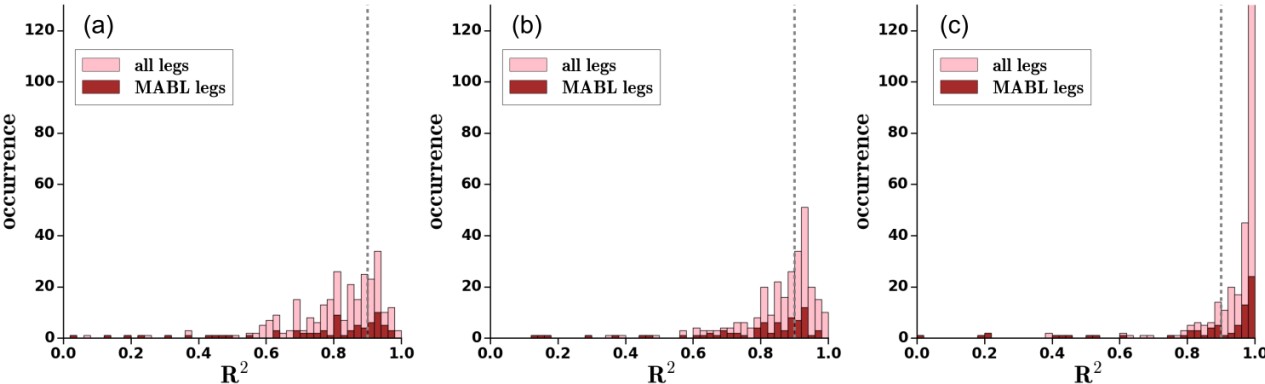

**Figure 4.** Histograms of the linear regression R-square ($R^2$) associated with the calibration of the KH20 fast humidity measurement with, as the reference slow measurement, (a) the 1011C miror hygrometer, (b) the WVSS2 sensor and (c) the WVSS2 sensor phased in time with the fast sensor. The comparison is done for all flights from RF11 to RF19, considering either the entire set of legs or just the legs flown within the MABL. The vertical dashed-line represents the $R^2 = 0.9$ threshold.

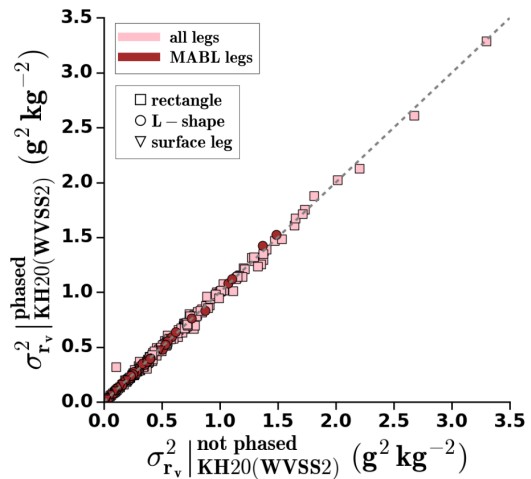

**Figure 5.** Variance of water vapor mixing ratio computed with the KH20 sensor, after calibration with the WVSS2 sensor phased in time, versus the variance obtained when the WVSS2 sensor is not phased. The comparison is done for all flights from RF11 to RF19, considering either the entire set of legs or just the legs flown within the MABL. The markers refer to the different types of leg, as sketched in Fig. 1.

The quality of the fast humidity measurements is thus assessed with respect to two metrics: $R^2$ and $RMSE$. For each sensor (KH20 or Licor), we define a green, yellow or red flag with respect to the combination of criteria on those two metrics (Fig. 6). The high quality (green) flag is defined by $R^2 \geq 0.9$ *and* $RMSE < 0.16$. At the opposite, the poor quality (red) flag is defined by $R^2 < 0.6$ *or* $RMSE > 0.6$. All other combinations of those two metrics correspond to an intermediate yellow flag. Note

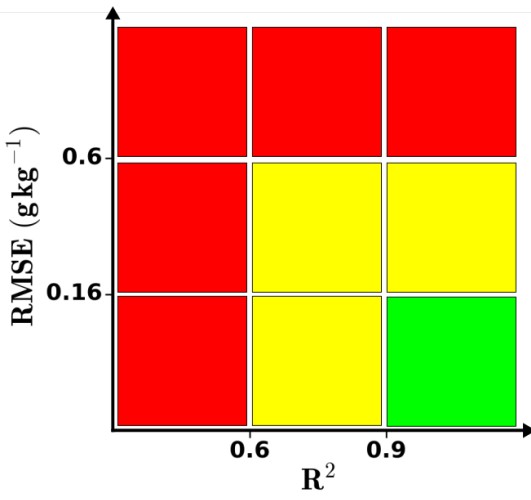

**Figure 6.** For each sensor (KH20 or Licor), the quality of high-rate humidity measurements is assessed against two metrics: $R^2$ (on the horizontal axis) and $RMSE$ (on the vertical axis). The quality decreases from green to red.

that the threshold values used to define these criteria result from a sensitivity analysis that compares the moisture flux and the variance obtained with the KH20 or the Licor sensors.

## 4.2 Comparison of the KH20 and Licor sensors

During EUREC[4]A, two fast sensors were mounted on the SAFIRE ATR 42: the Licor sensor, which had been previously adapted to the airplane, and the KH20 sensor, which was adapted to the airplane more recently in the hope of improving the performance of the high-rate humidity measurements.

Figure 7a shows an example of time series from both sensors during a subcloud layer segment of flight RF19, after the calibration process discussed previously. First, it shows that the signal from the Licor sensor was associated with a significant

noise. This feature was present along all the field campaign. Beside this noise issue, the Licor showed appropriate moisture measurements at lower frequencies, consistent with good R-square coefficients of the calibration ($R^2 = 0.99$ for both KH20 and Licor in Fig. 7). The corresponding spectra of those series shown in Fig. 7b exhibit more clearly the noise issue of the Licor. In contrast, the KH20 shows a nice behaviour of the spectra up to 6-8 Hz, notably showing the -2/3 slope in the inertial subrange. This means that this sensor can be used to study fine scale processes.

The determination of the thresholds of $R^2$ and $RMSE$ for the green flag introduced above was made such that the selected green-flagged legs showed a good consistency between the KH20 and the Licor on the estimates of moisture variance and covariance. This is illustrated with Fig. 8. Consistently, when we consider only the legs with green flags for both sensors, the agreement on variance (Fig. 8a) and moisture flux (Fig. 8c) is very good, especially relatively to the small intensity of turbulence found in EUREC[4]A, and the large associated random errors.

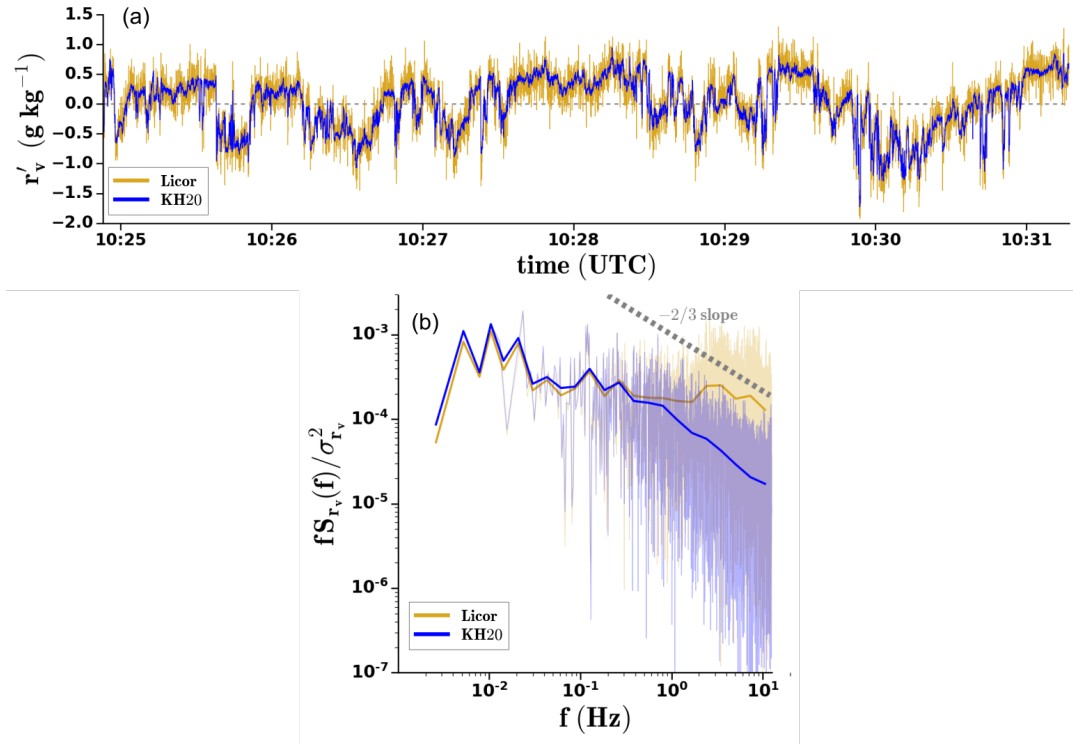

**Figure 7.** Comparison of Licor and KH20 signals on a leg flown at z 600 m on 13 February 2020 (RF19): (a) Time series of water vapour mixing ratio fluctuations from the Licor (in yellow) and from the KH20 (in blue). (b) Associated normalized energy density spectra. Smoothed spectra are indicated in thick solid lines.

The noise of the Licor signal naturally impacts the variance estimates, leading to an overestimation of about $0.05$ $\text{g}^2$ $\text{kg}^{-2}$ (Fig. 8a). However, the Licor noise does not significantly impact the covariance estimates of vertical velocity with moisture $(\overline{w'r'_v})$, as shown in Fig. 8c, because the noise signal is not correlated to the vertical velocity. Moreover, the energy of the correlation mainly ranges at scales larger than those where the noise predominates.

As a result of this analysis, the KH20 sensor is primarily used for turbulence moments estimates and analysis of the fine scale processes. But in case of strong failure of this sensor, the Licor is used as an alternative for the covariance estimates. The variances of the Licor were corrected for the noise (Fig. 8b), by using the value of the autocovariance function of moisture fluctuations at the fifth lag as an estimate of the variance. The use of the first lag is common and adapted for taking account of uncorrelated noise (Lenschow et al., 2000, 2012), since the autocovariance at zero lag is equal to the variance of the signal plus the variance of the white noise. Here, we found that using the fifth lag was more appropriate, due to slightly correlated noise, and the need to find a best compromise. This means that we lose the amplitude of the fluctuations of scales smaller than 20 m.

We found that generally, the KH20 sensor encountered issues in legs close to surface, due to sea salt. This is well shown by Fig. 8c, where the legs with green flag on Licor only (yellow-green) all show larger covariances with Licor. On the contrary,

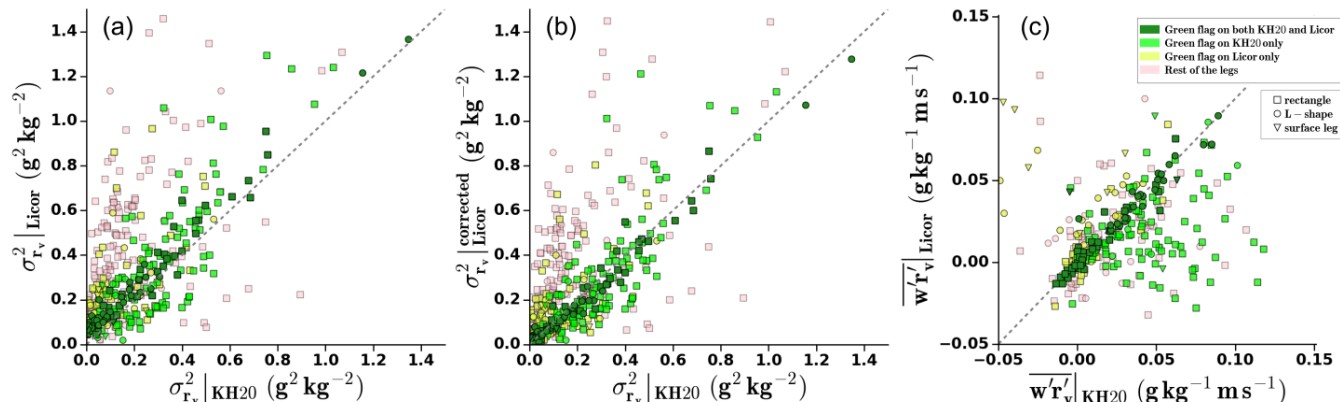

**Figure 8.** Humidity variances computed on 5-min segments for all the flights RF03 to RF19: (a) from the KH20 signals versus the Licor signals and (b) from the KH20 signals versus the Licor signal corrected for noise. (c) Moisture flux from the KH20 signal versus from the Licor signal for flights RF09 to RF19. The symbols correspond to the altitude of the leg. The dark green markers refer to legs with a good quality calibration of humidity for both sensors, bright green markers refer to legs with a good quality calibration of humidity for the KH20 sensor only, the yellow-green markers refer to legs with a good quality calibration of humidity for the Licor sensor only and finally, the bright pink markers correspond to the rest of the legs.

the Licor had more difficulties when the SAFIRE ATR 42 was crossing clouds or even rain during the 'R' legs, while the KH20 behaved much better in those wet conditions. Indeed, in Fig. 8c, all the legs associated with a green flag on KH20 only show
larger covariances with KH20 than for Licor.

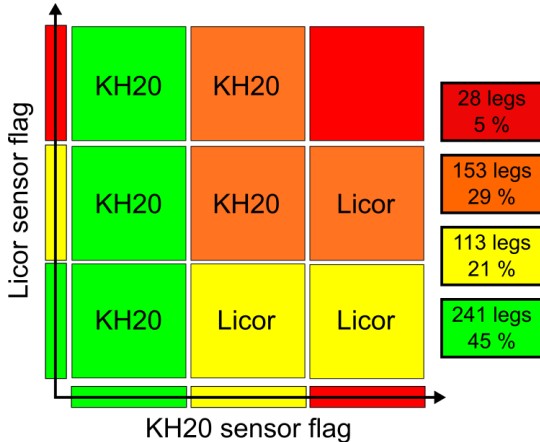

**Figure 9.** Definition of a combined flag for the humidity, depending of the KH20 and the Licor sensors quality flags. The number of segments and the associated percentage for each category of the combined flag is indicated in the boxes on the right. It refers to the flights RF03 to RF19 for the short legs.

In order to obtain the best estimate of the turbulent moments and fluctuations for each leg, they have been calculated using either on one sensor or the other, depending on their respective flags, with a priority given to the KH20 sensor. This results in the definition of a combined flag as illustrated in Fig. 9. In total, over the 535 five-minute segments, 241 segments are based on the KH20 with green flag (green combined flag), 113 are based on the Licor with green flag as an alternate (yellow combined flag), 153 on the KH20 or the Licor with yellow flag (orange combined flag), and 28 are unusable (red combined flag). Therefore, the calculation of the turbulent moments associated with humidity is trustworthy for the green and yellow combined flags. Orange flags should preferably be avoided and red flags are automatically invalidated. Finally, although the confidence in the calculation of turbulent moments for the yellow flag is good, the use of Licor fluctuations for studying fine scale processes, (e.g. with spectral analysis or probability density functions) should be avoided, as shown in Section 4.2. KH20 sensor should be preferred because of its better description of the expected spectrum in the inertial range, and its better recording of the amplitude and the distribution of the fluctuations.

## 5   Qualification of the fast temperature sensors

On board the SAFIRE ATR 42, the temperature was measured by two sensors: a Rosemount probe and a fine wire. The typical and reference Rosemount temperature probe showed some issues during the field campaign, including spurious negative spikes that were not visible on the fine wire sensor. Those were not easily explained, but supposed to be inherent to the sensor itself. Rarely, a large noise could also appear locally in the presence of cloud droplets. The housing of the Rosemount probe makes it difficult for the cloud droplets to penetrate into the probe and to reach the sensor. However, should a droplet reach the sensor, it takes more time to dry out. On the contrary, the fine wire is more exposed, but it recovers quickly. A usual weakness of the fine wire is its ability to break with shocks, in particular during take off or landing. During EUREC[4]A, the fine wires did not break, and turned out to provide a better fine scale signal than the Rosemount probe.

The two fine wires were installed starting at flight RF09, and calibrated with the Rosemount probe at 1Hz for each flight. Both fine wires were consistent together, but one showed some noise that the other did not show at all. We consider only the latter here. We considered this measurement as non-absolute, and used it only for the study of temperature fluctuations. We calibrated the fine wire with the raw impact temperature of the Rosemount probe temperature as a reference, with one calibration per flight. The regression slope was very close to 1 (1.07 in average, with a standard deviation of 1.2% over the 11 flights concerned). The most significant variability was found on the offset (coordinate at origin of the regression line), which varied between -4.6 and -1.9 °C , with a standard deviation of 2.6 °C. This variation may be explained by the fine wire resistance varying with time due to oxidation. From this calibration, and due to the incertitude of the housing features and recovery factor, we applied the same recovery factor of the Rosemount (0.98), to retrieve the static temperature from the impact temperature. Those results were similar to those found in the analysis of Baehr et al. (2002) on the same type of fine wire, and same antenna.

Figure 10a shows a time series of the temperature fluctuations derived from each sensor, during a subcloud layer leg of RF19. The spikes of the Rosemount temperature probe signal were particularly numerous in this example. The comparison also reveals

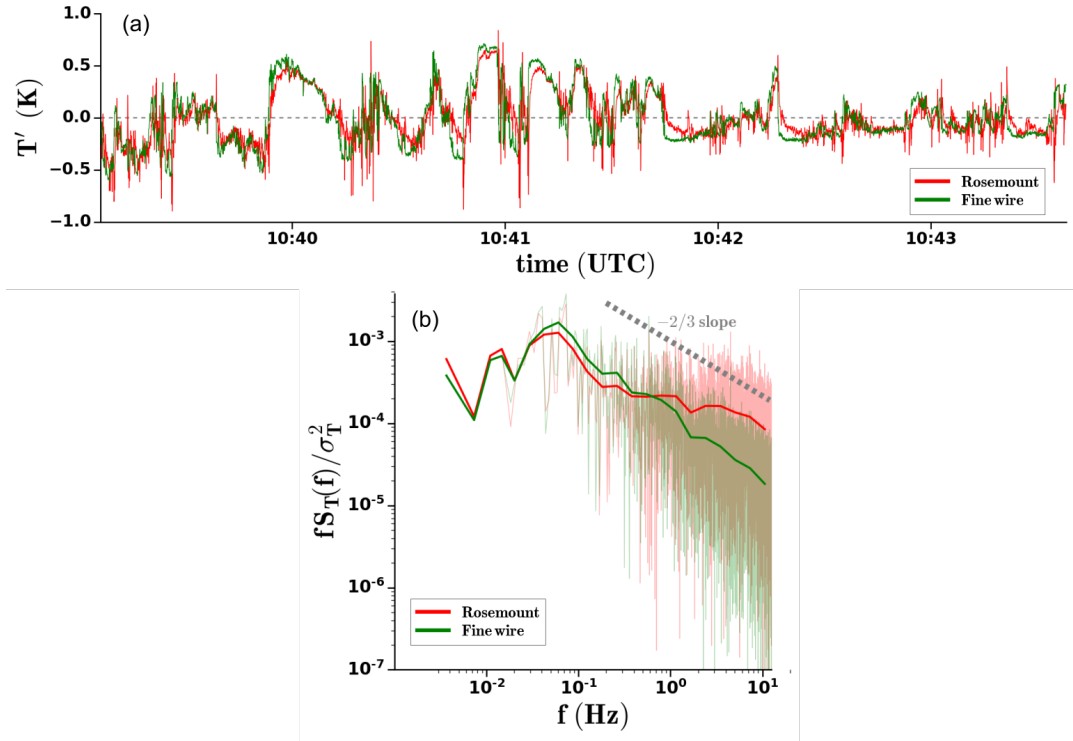

**Figure 10.** (a) Time series of the temperature fluctuations measured by the Rosemount probe (in red) and by the fine wire (in green) during a subcloud layer leg of RF19. (b) Corresponding density energy spectrum.

the shorter time response of the fine wire, and its better ability to catch the small-scale variability. This is confirmed by the
250 comparison of the spectra (in Fig. 10b), which shows how the fine wire temperature density energy spectrum (multiplied by the frequency) better follows the expected -2/3 slope in the inertial subrange.

The covariance between temperature and moisture is another evidence for the larger relevance of the fine wire signal (Fig. 11). Temperature and moisture fluctuations are often well correlated: for example, an intrusion of air from above is associated with a drier and warmer structure (negative and positive fluctuations of moisture and temperature, respectively).
Figure 11 shows that this correlation is higher when the temperature is measured by the fine wire than when it is measured by the Rosemount probe. It is partly explained by the fact that the temperature variance is larger with the fine wire, but also because the fine wire tracks the fine-scale fluctuations of temperature better than the Rosemount probe. For these reasons, the fine wire temperature signal was chosen during EUREC$^4$A for the best estimate of the turbulent moments and fluctuations. The Rosemount probe temperature was used as spare during the first part of the campaign, and during a few 'R' legs of RF17 and
RF19 where the fine wire sensor was heavily impacted by cloud droplets. A green flag was associated with the fine wire use, and a yellow flag with the use of the Rosemount probe.

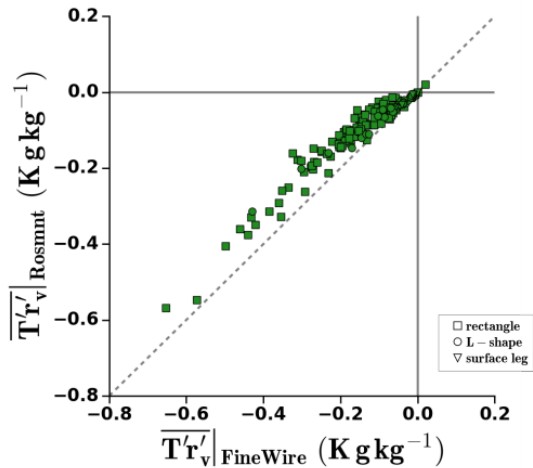

**Figure 11.** Comparison of the covariance of temperature and moisture obtained when using the Rosemount probe temperature (Y axis) and the fine wire temperature (X axis). In both cases, the calculation is based on the moisture fluctuations associated with a green flag.

## 6 Computation of turbulence moments and associated errors

After control and calibration, the 25 Hz fluctuations are used to compute the turbulence moments and other characteristics of the MABL turbulence. The turbulent moments or characteristics evaluated for each leg are listed in Table 2.

Only stabilized legs are considered for the turbulence data processing, due to the increase of errors on basic measurements during turns, or more generally during phases with varying flight attitude and speed. Moreover, to obtain an homogeneous statistical ensemble of turbulent moments associated with random and systematic error estimates, the straight horizontal legs are divided into segments of equal duration and length. Two types of segments are considered (Fig. 12): segments of 60 km / 10 min (referred to as 'longlegs'), which correspond to the length of a 'L' branch, and segments of 30 km / 5 min (referred to

as 'shortlegs'). As suggested by Lenschow et al. (1994), 30 km long segments are a good compromise, as they are long enough to sample the structures which dominate the turbulent exchanges and short enough to explore the spatial variability from one leg to the other. Note that the shortlegs are occasionally adjusted within the subcloud layer (during 'L' legs) to avoid water droplets (2 segments of that kind).

For each segment, the turbulent moments are calculated on two types of fluctuations time series: detrended series, or high-

275 pass filtered series, with a cut off frequency of 0.018 Hz (about 5 km wavelength). This filter is meant to remove the contribution of mesoscale features. The cut off wavelength is chosen based on the co-spectra of the vertical velocity with all other variables (temperature, humidity, horizontal components), so that all turbulent scales contributing to the covariance are taken into account.

Figure 13 shows an example of the filtered time series of five variables: $w'$, $\theta'$, $r'_v$, $u'_L$, $v'_T$. The prime symbol indicates

a fluctuation relatively to the mean value. Here $u'_L$ and $v'_T$ are respectively the longitudinal and lateral fluctuations of the horizontal wind relative to the mean wind over the considered leg. The longitudinal and lateral fluctuations of the horizontal

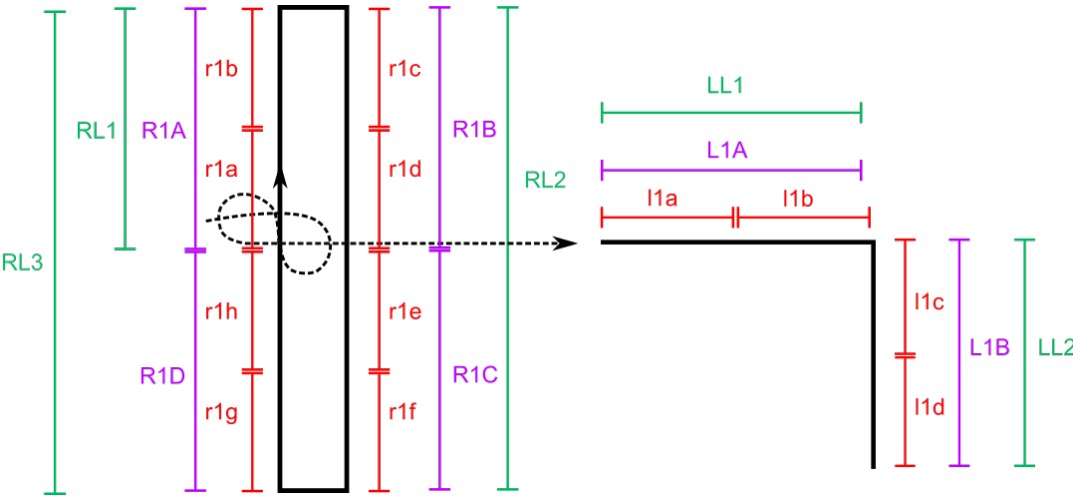

**Figure 12.** Schematic view of the segmentation of stabilized legs into segments of equal duration and length: 30 km / 5 min long segments ('shortlegs', in red), and 60 km / 10 min long segments ('longlegs', in purple). Also reported are the 'longestlegs' segments (in green), which are the longest stabilized segments in one direction.

wind relative to the aircraft, $u'_x$, $v'_y$, are also calculated and made available in the dataset, as well as the fluctuations of eastward and northward components. The use of one or the other referential depends on the purpose of the turbulence data analysis. In all three referentials, the vertical velocity is taken positive upward, and the referential systems are direct and orthogonal.

The second and third order turbulent moments are computed with the eddy correlation method. The covariance of two variables $x$ and $y$ is defined as:

$$\overline{x'y'} = \frac{1}{T} \int_0^T x'(t)y'(t)dt, \tag{1}$$

where $T$ is the duration of the leg. From the variances of $u$, $v$, and $w$, the Turbulent Kinetic Energy (TKE) is calculated as $TKE = \frac{1}{2}(\sigma_u^2 + \sigma_v^2 + \sigma_w^2)$. Third-order turbulent moments are also computed, enabling the calculation of the skewness of a

variable $x$:

$$S_x = \frac{\overline{x'^3}}{\left(\overline{x'^2}\right)^{3/2}}. \tag{2}$$

For both second-order and third-order moments, the ratio, noted '$R$' in Table 2, between the moment obtained without high-pass filtering (fluctuations obtained only by detrending the original series) and that obtained after high-pass filtering is computed. This index informs about the stationarity and homogeneity of a sample. For a perfectly homogeneous and stationary sample,

with no impact of meso-scale or sub-mesoscale structures, this ratio should theoretically be equal to 1. It is often close to one for vertical velocity variance, but can be much larger for other variables, and for covariances.

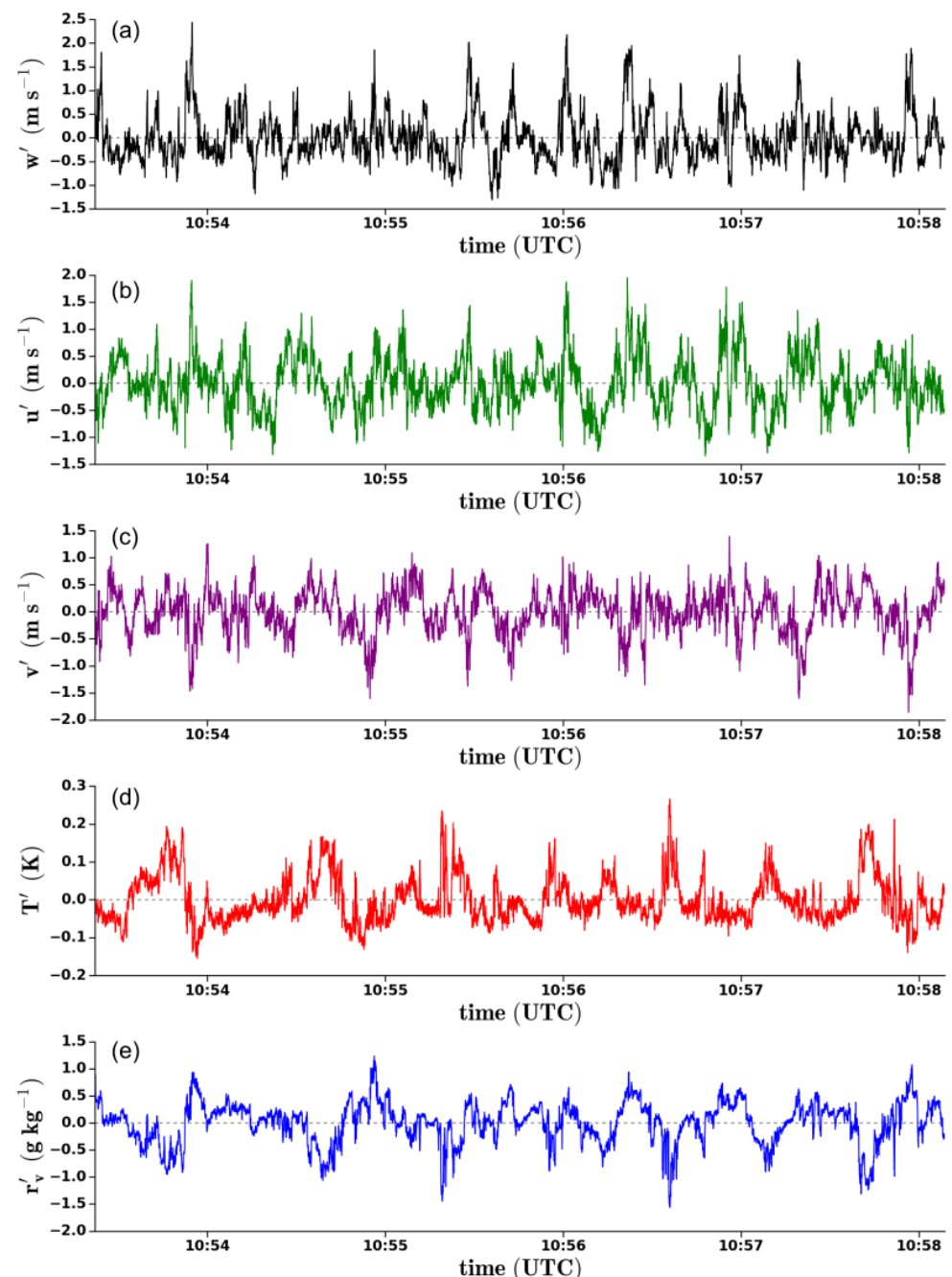

**Figure 13.** Time series of the filtered fluctuations of (a) vertical velocity, (b) wind velocity component longitudinal to the wind (c) wind velocity component transverse to the wind (d) potential temperature and (e) water vapour mixing ratio during a leg flown at z~275 m on 13 February 2020 (RF19).

Characteristic lengthscales suitable to describe the turbulence field are also computed, such as the wavelength of the vertical velocity spectrum peak or integral lengthscales. The lengthscale of the maximum spectral density energy of vertical velocity is obtained by fitting an analytical spectrum of the form $fS(f) = \frac{fS_0}{1+1.5\left(\frac{f}{f_0}\right)^{5/3}}$, where $f$ is the frequency, and $S_0$ and $f_0$ are fitted to the observed vertical velocity spectra (Lambert and Durand, 1999; Attie et al., 1999). Depending on purpose, more complex analytical spectra may be used to estimate the wavelength of maximum spectral energy (see e. g. Kristensen et al.,1989; Lothon et al., 2009 or Brilouet et al., 2017) and other definitions (e. g. Pino et al., 2006). This one is chosen for the sake of simplicity.

The integral lengthscale of a variable $x$ is estimated as the integral of the normalized autocorrelation function from zero lag ($\tau = 0$) to the first zero ($\tau_0$) of the function (Lenschow et al., 1994):

$$\mathcal{L}_x = \frac{1}{\overline{x'^2}} \int_0^{\tau_0} \overline{x'(t)x'(t+\tau)} d\tau. \tag{3}$$

The turbulent kinetic energy dissipation rate ($\varepsilon$) is estimated from the vertical velocity energy spectrum $S_w$ in the inertial subrange (Lambert and Durand, 1998), based on the Kolmogorov formulations: $S_w(k) = \frac{4}{3}\alpha\varepsilon^{2/3}k^{-5/3}$, where $k$ is the wavenumber ($k = \frac{2\pi}{TAS}f$) and $\alpha$ is the Kolmogorov constant, taken as $\alpha=0.52$ (Fairall and Larsen, 1986).

The reliability and accuracy of the observed turbulent moment estimates can be assessed based on sampling and filtering conditions. As introduced by Lenschow et al. (1994), using high-pass-filtered and finite-length samples generates an error which can be decomposed into two contributions: a systematic error ($\epsilon_s$) and a random error ($\epsilon_r$). The systematic error reflects the loss of information due to the high-pass filtering and can be estimated as the difference between the covariance of the detrended series ($F_{det}$) and the covariance of the high-pass filtered series ($F_{fil}$):

$$\epsilon_s = \frac{F_{det} - F_{fil}}{F_{det}}. \tag{4}$$

The random error is generated by the finite length of the sample and is therefore inherent in the measurement and can not be removed. For a covariance between two variables $x$ and $y$, the associated random error can be estimated by:

$$\epsilon_r = \sqrt{\frac{2\mathcal{L}_{xy}}{L}}\sqrt{1 + \frac{1}{r_{xy}^2}}, \tag{5}$$

where $\mathcal{L}_{xy}$ is the integral lengthscale of $x'y'$, $L$ the length of the leg and $r_{xy}$ the correlation coefficient between $x$ and $y$.

**Table 2.** List of turbulent parameters calculated over each segment, their standard nomenclature and their names in the netcdf files.

| | | | | |
|---|---|---|---|---|
| **General characteristics** | | | | |
| Start, end, central time | $t_i, t_f, \overline{t}$ | | | 'time_start', 'time_end', 'time' |
| Start, end, central position | $lat_i, lon_i, lat_f, lon_f, \overline{lat}, \overline{lon}$ | | | 'lat_start', 'lon_start', 'lat_end', 'lon_end', 'lat', 'lon' |
| duration | $T$ | | | 'duration' |
| Mean heading | $\overline{THDG}$ | | | 'MEAN_THDG' |
| Mean true airspeed | $\overline{TAS}$ | | | 'MEAN_TAS' |
| Mean ground velocity | $\overline{GS}$ | | | 'MEAN_GS' |
| Mean height above the sea | $\overline{z}$ | | | 'alt' |
| Mean pressure | $\overline{P}$ | | | 'MEAN_P' |
| Mean static temperature | $\overline{T_s}$ | | | 'MEAN_TS' |
| Mean air density | $\overline{\rho_a}$ | | | 'MEAN_RHO_A' |
| **Calibration informations** | | | | |
| Quality flag for humidity signal | | | | 'QC_MR' |
| Humidity sensor used | | | | 'HUM_SENSOR' |
| Quality flag for temperature signal | | | | 'QC_T' |
| Temperature sensor | | | | 'TEMP_SENSOR' |
| **First order moments** | | | | |
| Mean wind | $\overline{FF}, \overline{DD}$ | | | 'MEAN_WSPD', 'MEAN_WDIR' |
| Mean potential temperature | $\overline{\theta}$ | | | 'MEAN_THETA' |
| Mean water vapour mixing ratio | $\overline{r_v}$ | | | 'MEAN_MR' |
| **Second order moments, filtering ratios and associated errors** | | | | |
| | $\overline{u'^2}$ | $R_{\overline{u'^2}}$ | $\epsilon_{s_{\overline{u'^2}}}$ | 'VAR_U', 'RATIO_VAR_U', 'ERR_S_VAR_U' |
| Wind components variance | $\overline{v'^2}$ | $R_{\overline{v'^2}}$ | $\epsilon_{s_{\overline{v'^2}}}$ | 'VAR_V', 'RATIO_VAR_V', 'ERR_S_VAR_V' |
| | $\overline{w'^2}$ | $R_{\overline{w'^2}}$ | $\epsilon_{s_{\overline{w'^2}}}$ | 'VAR_W', 'RATIO_VAR_W', 'ERR_S_VAR_W' |
| Turbulent kinetic energy | $e$ | $R_e$ | | 'TKE', 'RATIO_TKE' |
| Turbulent kinetic energy dissipation rate | $\varepsilon$ | | | 'EPSILON_TKE' |
| Potential temperature variance | $\overline{\theta'^2}$ | $R_{\overline{\theta'^2}}$ | $\epsilon_{s_{\overline{\theta'^2}}}$ | 'VAR_T', 'RATIO_VAR_T', 'ERR_S_VAR_T' |
| Water vapour mixing ratio variance | $\overline{r_v'^2}$ | $R_{\overline{r_v'^2}}$ | $\epsilon_{s_{\overline{r_v'^2}}}$ | 'VAR_MR', 'RATIO_VAR_MR', 'ERR_S_VAR_MR' |
| | $\overline{w'u'}$ | $R_{\overline{w'u'}}$ | $\epsilon_{s_{\overline{w'u'}}}$ $\epsilon_{r_{\overline{w'u'}}}$ | 'COVAR_WU', 'RATIO_COVAR_WU', 'ERR_S_COVAR_WU', 'ERR_R_COVAR_WU' |
| Covariances with the vertical velocity | $\overline{w'v'}$ | $R_{\overline{w'v'}}$ | $\epsilon_{s_{\overline{w'v'}}}$ $\epsilon_{r_{\overline{w'v'}}}$ | 'COVAR_WV', 'RATIO_COVAR_WV', 'ERR_S_COVAR_WV', 'ERR_R_COVAR_WV' |
| | $\overline{w'\theta'}$ | $R_{\overline{w'\theta'}}$ | $\epsilon_{s_{\overline{w'\theta'}}}$ $\epsilon_{r_{\overline{w'\theta'}}}$ | 'COVAR_WT', 'RATIO_COVAR_WT', 'ERR_S_COVAR_WT', 'ERR_R_COVAR_WT' |
| | $\overline{w'r_v'}$ | $R_{\overline{w'r_v'}}$ | $\epsilon_{s_{\overline{w'r_v'}}}$ $\epsilon_{r_{\overline{w'r_v'}}}$ | 'COVAR_WMR', 'RATIO_COVAR_WMR', 'ERR_S_COVAR_WMR', 'ERR_R_COVAR_WMR' |
| **Third order moments and filtering ratios** | | | | |
| | $\overline{u'^3}$ | $R_{\overline{u'^3}}$ | | 'M3_U', 'RATIO_M3_U' |
| Wind component third order moment | $\overline{v'^3}$ | $R_{\overline{v'^3}}$ | | 'M3_V', 'RATIO_M3_V' |
| | $\overline{w'^3}$ | $R_{\overline{w'^3}}$ | | 'M3_W', 'RATIO_M3_W' |
| Potential temperature third order moment | $\overline{\theta'^3}$ | $R_{\overline{\theta'^3}}$ | | 'M3_THETA', 'RATIO_M3_THETA' |
| Water vapour mixing ratio third order moment | $\overline{r_v'^3}$ | $R_{\overline{r_v'^3}}$ | | 'M3_MR', 'RATIO_M3_MR' |
| Skewness of each thermodynamic variables | $S_u, S_v, S_w, S_\theta, S_{r_v}$ | | | 'SKEW_U', 'SKEW_V', 'SKEW_W', 'SKEW_T', 'SKEW_MR' |
| **Characteristic lengthscales** | | | | |
| Vertical velocity spectrum peak wavelength | $\lambda_w$ | | | 'LAMBDA_W' |
| Integral lengthscales | $\mathcal{L}_w, \mathcal{L}_{wu}, \mathcal{L}_{wv}, \mathcal{L}_{w\theta}, \mathcal{L}_{wr_v}$ | | | 'L_W', 'L_WU', 'L_WV', 'L_WT', 'L_WMR' |

## 7 Available dataset

The dataset includes two kinds of data:

1. 'Moments': the turbulent moments calculated over each segment. The data are stored in NetCDF files (with one file per flight) for the three sets of short (30 km), long (60 km) segments and over the longest possible segments of stabilized legs. However, one should be careful that these 'longest' segments have different lengths, that range from 60 to 125 km. .

2. 'Fluctuations': the time series of fluctuations over each segment, filtered and detrended only, are made available to enable specific analyses or estimates of the turbulent moments through an alternative approach. There is one NetCDF file per segment and per flight.

Note that the fluctuations and moments over the longest possible segments are also made available, even if this last set is composed of segments of different lengths (from 60 to 125 km). It enables any user to work on the longest series of calibrated 330 fluctuations, for the entire stabilized legs. Of course, moments of 'R' legs of 125 km are likely still more heterogeneous, and should be considered only in specific strict conditions.

For both the turbulent fluctuations and turbulent moments, two levels of data processing are considered:

– Level 2: all files of turbulent moments and fluctuations, calculated for each sensor of temperature and humidity.

– Level 3: 'best estimates': turbulent moments and fluctuations derived from the sensors (or their combination) that have 335 the best quality flags for the leg under consideration (the flags are described in sections 4 and 5); for each leg, they are considered as the best estimates of moments and fluctuations given the available instrumentation.

**Table 3.** File nomenclature. *YYYYMMDD* corresponds to the flight date (e. g. '20200202'), *NN* to the flight number (e. g. '09'), *LEG* to the segment identifier (e. g. 'R2B' or 'l2c'), and *LEVEL* to the level of data processing ('L2' or 'L3').

| Moments | |
|---|---|
| 30 km segments | EUREC4A_ATR_turbulence_moments_*YYYYMMDD*_RF*NN*_shortlegs_*LEVEL_version*.nc |
| 60 km segments | EUREC4A_ATR_turbulence_moments_*YYYYMMDD*_RF*NN*_longlegs_*LEVEL_version*.nc |
| Longest segments | EUREC4A_ATR_turbulence_moments_*YYYYMMDD*_RF*NN*_longestlegs_*LEVEL_version*.nc |
| Fluctuations | |
| 30 km segments | EUREC4A_ATR_turbulence_fluctuations_*YYYYYMMDD*_RF*NN*_*LEG_LEVEL_version*.nc |
| 60 km segments | EUREC4A_ATR_turbulence_fluctuations_*YYYYYMMDD*_RF*NN*_*LEG_LEVEL_version*.nc |
| Longest segments | EUREC4A_ATR_turbulence_fluctuations_*YYYYYMMDD*_RF*NN*_*LEG_LEVEL_version*.nc |

Table 3 explains the file nomenclature, using Fig. 12 for the naming of each segment. For each flight, a 'yaml' file is provided together with the dataset, that defines the start and end times of each flight segment.

| RF | Date | $u$ | $v$ | $w$ | $T$ | $r_v$ |
|----|------|-----|-----|-----|-----|-------|
| 03 | 26/01 | | | | | |
| 04 | 26/01 | | | | | |
| 05 | 28/01 | | | | | |
| 06 | 30/01 | | | | | |
| 07 | 31/01 | | | | | |
| 08 | 31/01 | | | | | |
| 09 | 02/02 | | | | | |
| 10 | 02/02 | | | | | |
| 11 | 05/02 | | | | | |
| 12 | 05/02 | | | | | |
| 13 | 07/02 | | | | | |
| 14 | 07/02 | | | | | |
| 15 | 09/02 | | | | | |
| 16 | 09/02 | | | | | |
| 17 | 11/02 | | | | | |
| 18 | 11/02 | | | | | |
| 19 | 13/02 | | | | | |

**Figure 14.** Quality flags associated with the turbulent data of each SAFIRE ATR 42 flight during the campaign. The data quality increases from red (poor quality) to orange, yellow and green (high quality). For moisture and temperature, the flag is the combined flag described in sections 4 and 5, considering the short legs of L3 data.

Figure 14 summarizes the availability and the quality of the high-rate data associated with each key variable. Based on the quality control described in sections 5 and 6 for temperature and moisture, it displays the proportion of legs within each flight that are associated with a green flag, as established on the basis of the L3 'shortlegs' dataset. This table shows how the quality of the measurements improved in the second phase of the field (from RF09), with the best quality and availability achieved from RF15 onwards.

As an illustration of the dataset, Fig. 15 shows an overview of the vertical profiles of variances for the entire field experiment (RF03 to RF19). The profiles are normalized by the lifting condensation level ($LCL$), estimated here as the flight altitude of the rectangle at the cloud base minus 50 m. Overall, the turbulence is weak in the subcloud layer during EUREC[4]A. As expected in a MABL, the variance of vertical velocity is maximum within the first half of the subcloud layer, and the variance of the horizontal wind is larger near the sea surface. The variances of temperature and moisture are maximum near cloud base and the entrainment layer, and they are minimum close to the surface. The very large scatter of the turbulent moments on 'R' legs flown around cloud-base likely reflects the large heterogeneity of the samples, related to the crossing of clouds and the mix of airmasses of very different origins and characteristics (including non turbulent airmasses). Over these legs, the moments need to be taken with much caution because their definition assumes an homogeneous sample, an hypothesis which is rarely met at cloud base.

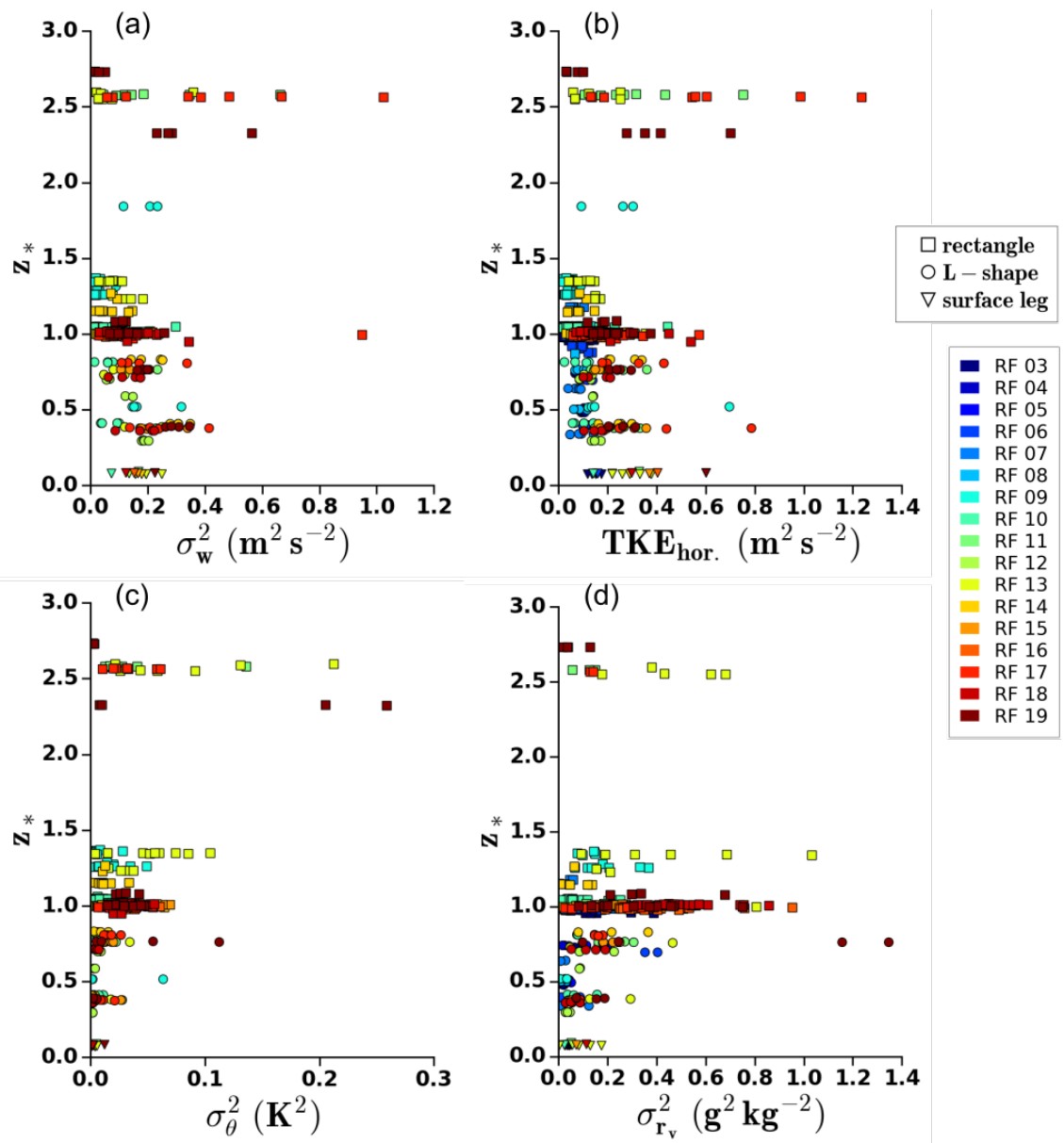

**Figure 15.** Normalized vertical profiles of variance of (a) vertical velocity, (b) horizontal turbulent kinetic energy, (c) temperature and (d) water vapour mixing ratio. Flight numbers are indicated in the top right box. For the water vapour mixing ratio, only the legs with a green or a yellow combined flag have been considered. The normalized altitude $z_*$ is defined by $z/LCL$ with $LCL$, the lifting condensation level.

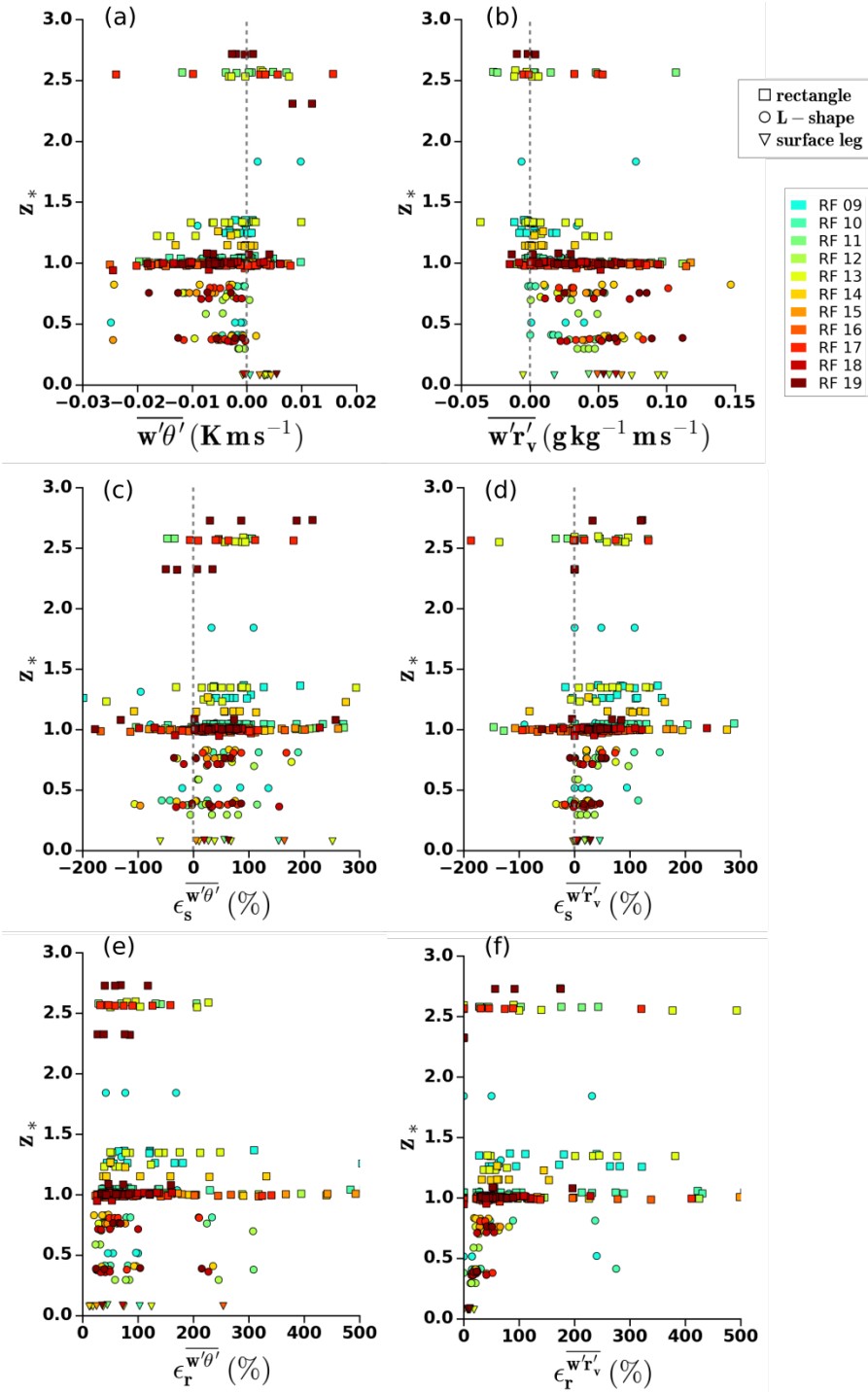

**Figure 16.** Normalized vertical profiles of (a) the heat flux, (b) the moisture flux, systematic error (c) for the heat flux, (d) for the moisture flux, random error (e) for the heat flux and (f) for the moisture flux. Flight numbers are indicated in the top right box. The normalized altitude $z_*$ is defined by $z/LCL$ with $LCL$, the lifting condensation level.

Finally, Fig. 16 shows, for the flights RF09 to RF19, the vertical profiles of covariance of the vertical velocity with temperature (Fig. 16a) or moisture (Fig. 16b), as well as their corresponding systematic and random errors. Around cloud base ($z_* \sim 1$), the random and systematic errors are very large. This is explained by the large heterogeneity of the samples at this level, and indicates that the vertical flux estimates are mostly relevant below cloud base.

The sensible heat flux is very small near the surface (likely due to the small air-sea temperature difference), and changes sign with height, consistent with the entrainment near cloud base. The moisture flux is more significant, with a large value near the surface, and an overall decrease of the flux with height.

For both the heat flux and the moisture flux, the systematic error can be particularly large. This is largely due the small fluctuations observed, resulting in very weak fluxes. Inside the MABL, the random error increases with altitude, partly related to the growth of the turbulent eddies. The profiles are similar to those found by Brilouet et al. (2017) during the HYMEX campaign over the Mediterranean sea, which took place in much stronger wind conditions than here though. Like in HYMEX, we find larger errors on the heat flux than on the moisture flux, due to the more significant moisture fluctuations.

On the 'R' legs above the MABL, the errors display a wide variability and potentially large values. This again reflects the high heterogeneity of the samples and the influence of the mesoscales at this hight.

## 8    Conclusions

This paper presents the EUREC[4]A turbulent dataset that has been produced based on the high rate in situ measurements of wind, temperature and moisture from the SAFIRE ATR 42. It explains the data processing strategies, the calibration methodologies, the procedures of quality control applied to the 25 Hz temperature and moisture measurements, and the methods used to estimate the turbulent moments and their associated errors.

The redundancy of temperature and moisture sensors on board the aircraft enabled us to overcome the failure of one or the other sensor, and to optimize the data processing. All turbulent moments and time series of turbulent fluctuations are associated with some information about the sensor(s) from which they are derived, plus a quality flag. These data constitute the Level-2 dataset. In addition, a Level-3 dataset provides an ensemble of 'best estimates' of the turbulent moments and fluctuations over each stabilized leg of the SAFIRE ATR 42 flights.

Considering our analysis of the data, and the flight strategy and conditions, we make the following remarks and recommendations to the future users of this dataset:

- The data collected at cloud-base over the 'R' legs or segments should be used with great caution. First, the presence of cloud droplets or rain may affect the performance of the high rate sensors. In addition, at this level the aircraft probes very contrasted air masses, including clouds and cloud-free air originating from the subcloud layer or entrained from above. The large heterogeneity of the samples makes the calculation of turbulent moments quite uncertain around cloud base. However, the turbulent fluctuations remain relevant and can be used for specific analyses such as conditional sampling, object approaches or case studies.

- The moisture fluctuations measured by the Licor sensor, and the temperature fluctuations measured by the Rosemount probe, exhibit limitations at very fine scales. The variance and covariance estimates are not affected by these limitations, but we recommend that the spectral or distribution analyses of the turbulent fluctuations primarily use the data from the KH20 moisture sensor and from the fine wire temperature sensor.

- Owing to the weakness of the turbulent fluxes during EUREC$^4$A, the turbulent moments estimates are associated with large systematic and random errors. This, added to the limited vertical sampling of the MABL, suggests that extrapolating sea surface turbulent fluxes from this dataset would not be accurate.

Despite these issues and the technical difficulties encountered at the beginning of the campaign, a rich and quality-controlled dataset has been produced based on the high rate measurements of the SAFIRE ATR 42, that will make it possible to study the
395 turbulence of the MABL during EUREC$^4$A.

These data will be used to characterize the structure and the variability of the subcloud layer, and the level of organization encountered underneath the clouds. Used jointly with the other EUREC$^4$A datasets from aircraft, balloons or unmanned aerial vehicles, they will help to decipher the nature of clouds-circulation interactions and to identify the roots of the shallow convective organization. They will also help evaluate the ability of large-eddy simulations to predict the characteristics of turbulence
within the subcloud layer of trade-wind regimes for a range of large-scale conditions.

*Data availability.* The dataset is available at https://eurec4a.aeris-data.fr/, (https://doi.org/10.25326/128 ; Brilouet et al., 2020).

*Author contributions.* TEXT

Pierre-Etienne Brilouet and Marie Lothon designed the data processing strategy, processed and analyzed the data and wrote the manuscript. They participated to the SAFIRE ATR 42 flight coordination onboard during the field experiment.
Jean-Claude Etienne and Pascal Richard processed the initial SAFIRE data to generate the 25 Hz data and participated to the data quality control.

Julien Lernoult was the SAFIRE instrumentalist mostly involved in the adaptation of the KH20 sensor, he monitored and maintained the sensors along the field experiment.

Hubert Bellec, Gilles Vergez, Thierry Perrin, Julien Lernoult are instrumentalist experts on the SAFIRE ATR 42 who con-
410 tributed to the preparation of the field and the field campaign itself, as regard the airborne instrumentation.

Tetyana Jiang, Frédéric Pouvesle, Claude Lainard, Michel Cluzeau were responsible for the data acquisition onboaord, and the preliminary data processing.

Patrice Medina, Theotime Charoy and Laurent Giraud contributed with SAFIRE instrumentalists to the initial adaptation of the KH20 sensor to the aircraft at the very start of the instrumental project.

Sandrine Bony is the co-lead coordinator (with Bjorn Stevens) of the EUREC$^4$A campaign, and the lead scientific coordinator of the SAFIRE ATR operations. She participated in the flights and in the monitoring of the measurements. She participated in the discussions on data analysis, and edited the manuscript.

Julien Delanoë and Marie Lothon were co-coordinator of the SAFIRE ATR 42 operations during the EUREC$^4$A campaign.

*Competing interests.* TEXT

*Acknowledgements.* Airborne data were obtained from the ATR aircraft operated by SAFIRE, the French facility for airborne research, an infrastructure of the French National Center for Scientific Research (CNRS), Météo-France and the French National Center for Space Studies (CNES). Distributed data are processed by SAFIRE and CNRM/GMEI/TRAMM. The authors gratefully acknowledge all the SAFIRE staff, technicians, engineers, pilots and directors, for their considerable efforts and involvement in the realization of the EUREC$^4$A airborne operations. We also thank Pierre Durand (Laboratoire d'Aérologie, University of Toulouse, UPS, CNRS, France) for his contribution to the
development of the KH20 aircraft-adapted sensor at its origin, as first initiative. We thank the Caribbean Regional Security System (RSS) for hosting the ATR and the ATR team in Barbados during the experiment; David Farrell and the Caribbean Institute for Meteorology and Hydrology (CIMH) for their logistical and administrative support; and the Department of Civil Aviation in Barbados and Andrea Hausold (from DLR), for their help and support of airborne operations. The authors also thank AERIS for their support during the campaign and for managing the EUREC$^4$A database. We also warmly thank the ground-based support team, especially Raphaela Vogel and Jessica Vial
from LMD, for there contribution to the real-time flight strategy, and Cyrille Flamant for his contribution to the coordination of the flight operations. Finally we thank Bjorn Stevens for co-coordinating EUREC$^4$A with Sandrine Bony in a professional, ingineous and enthousiastic way. The EUREC$^4$A project was supported by the European Research Council (ERC) under the European Union's Horizon 2020 research and innovation programme (grant agreement no. 694768) with some additional support from the French Space Agency CNES through the EECLAT proposal.

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
