# Peer review of "The EUREC4A turbulence dataset derived from the SAFIRE ATR 42 aircraft"

_Earth System Science Data, 2021_

## Referee Comment (RC1)

Review of Brilouet et al: "The EUREC$^4$A turbulence dataset derived from the SAFIRE ATR 42 aircraft"

Submitted to *Earth System Science Data (ESSD)*

Summary Comment-
This paper describes the derived/processed turbulence parameters computed from the high-rate (~25 Hz) dataset collected the SAFIRE ATR 42 aircraft during EUREC$^4$A. The manuscript is well-written and easy to follow. The data set are accessible as described. A perusal of the data files indicates the data appear to be complete and follow the description as laid out in the manuscript. The data set is suitable for publication with ESSD as I see potential for future use of the data set for a range of atmospheric scientists. For these reasons, I recommend publication once the authors address a few comments below.

Specific Comment-
1. I think the manuscript would benefit from a bit more additional information in the introduction or in section 2. Specifically, one to two paragraphs describing the data set would have helped me as I read through the manuscript. Little information about the data set itself was provided until Section 7—I think a bit more information upfront about a general description would be useful. The very specific information could still be retained later in the manuscript (Section 7).
2. The authors provide a very good, detailed description of the handling of the humidity data/measurements. In a revised version, the authors should include more information on the handling of the temperature measurements. I view this as the one major lacking component of the manuscript/data handling description. The Rosemount (Total temperature) housing is notoriously susceptible to wetting of the element—the authors point to this, but don't really discuss what impacts wetting has their measurements nor how they identify wetting in the data set. No information is provided about the fine-wire, not even a reference is provided for this sensor. Some description of the sensor itself should be provided—is it housed? Fully open to the free-stream? If the latter, how is the sensor protected against radiative effects? How is the recovery factor (of the element and the housing) determined and accounted for?

Technical/Minor Comments-
1. Line 7 (abstract) delete 'a fast rate' and simply replace with '25 Hz'
2. Line 45 'Section 2 describes…'
3. Figure 1 – in caption, note that R-pattern is shown in red and L-pattern is shown in blue. Remove reference to S-Pattern. The last sentence in the figure description does not make sense to me…I'm not sure what the authors are trying to convey.
4. Table 1 – The authors should provide some description about the shorthand being used—it took me a while to figure out that, for example, $R_{strati}$(1830 m) + 2$R_{cb}$ (680m – 740 m) referred to "1 R pattern in stratiform clouds at 1830 m and 2 R patterns at cloud base, one at 680 m and the other at 740 m" --- I still don't know what $L_{flower}$ is?

5. Line 87-88 – Inertial navigation unit (Xsea model)? Is this a manufacturer and model number? I've never heard of that and didn't find anything with a quick internet search.
6. Line 126 – Krypton
7. Line 142 – KH20 showed 'a very good behavior.' -- What do you mean by this? It tracked well with other measures? Authors need to be more descriptive here.
8. Line 268 – orthogonal? (not orthonormed…)

---

## Author Comment (AC1)

**Answer to reviewer # 1**

"The EUREC⁴A turbulence dataset derived from the SAFIRE ATR 42 aircraft"

Pierre-Etienne Brilouet, Marie Lothon, Jean-Claude Etienne, Pascal Richard, Sandrine Bony, Julien Lernoult, Hubert Bellec, Gilles Vergez, Thierry Perrin, Julien Delanoë, Tetyana Jiang, Frédéric Pouvesle, Claude Lainardn, Michel Cluzeau, Laurent Guiraud, Patrice Medina, Theotime Charoy

**The authors would like to thank the anonymous reviewer for his/her suggestions and relevant remarks, which helped us to improve the manuscript. The original text from the review is written in black below, our reply in blue and the proposed modifications of the manuscript in red.**
* * *
**Summary Comment -**

This paper describes the derived/processed turbulence parameters computed from the high-rate (~25 Hz) dataset collected the SAFIRE ATR 42 aircraft during EUREC 4 A. The manuscript is well-written and easy to follow. The data set are accessible as described. A perusal of the data files indicates the data appear to be complete and follow the description as laid out in the manuscript. The data set is suitable for publication with ESSD as I see potential for future use of the data set for a range of atmospheric scientists. For these reasons, I recommend publication once the authors address a few comments below.

**Specific Comment -**

1. I think the manuscript would benefit from a bit more additional information in the introduction or in section 2. Specifically, one to two paragraphs describing the data set would have helped me as I read through the manuscript. Little information about the data set itself was provided until Section 7 —I think a bit more information upfront about a general description would be useful. The very specific information could still be retained later in the manuscript (Section 7).

We understand this point, and the need to very shortly describe the dataset, before it is fully detailed in sections 6 and 7. As suggested, a general description of the dataset has been added in the end of the introduction, before the outline. Since the description is included here before the details about the data-processing, we find appropriate to remain concise. The complete information on the data set is given in section 7. The following clarifications have been added in the text:

*"This paper describes the EUREC4A dataset containing the turbulent fluctuations and turbulent moments associated with the high frequency measurements of temperature, moisture and wind from the SAFIRE ATR 42 aircraft, computed over horizontal stabilized legs."*

2. The authors provide a very good, detailed description of the handling of the humidity data/measurements. In a revised version, the authors should include more information on the handling of the temperature measurements. I view this as the one major lacking component of the manuscript/data handling description. The Rosemount (Total temperature) housing is notoriously susceptible to wetting of the element—the authors point to this, but don't really discuss what impacts wetting has their measurements nor how they identify wetting in the data set. No information is provided about the fine-wire, not even a reference is provided for this sensor. Some description of the sensor itself should be provided—is it housed? Fully open to the free-stream? If the latter, how is the sensor protected against radiative effects? How is the recovery factor (of the element and the housing) determined and accounted for?

It is true that the description of the temperature was short in the manuscript, and that more details should be given.

The Rosemount probe wetting is actually not easily detectable, among other aspects like salting and some other technical issues, more intrinsic to the sensor itself, and which manifested as some spurious noise, and numerous spikes for some flights. Those, however, did not significantly impact the slow rate (1 Hz) of the temperature measurement. We think that the sensor itself had some unexpected issues during EUREC4A, mostly independent of the presence of droplets.

The fine wire sensor used in EUREC4A is an home-made sensor, analyzed in Baehr et al. 2002. However, there is no specific reference related to this sensor, other than this internal report. The two platinum fine wires are housed in a tubular antenna from SFIM company (model T4113). They are directly exposed to the stream, but protected from radiation, which consequently should not have a significant impact.
We considered this measurement as non-absolute, and used it only for the study of temperature fluctuations. We calibrated the fine wire with the raw impact temperature of the Rosemount probe temperature as a reference, with one calibration per flight. The regression slope was very close to 1 (1.07 in average, with a standard deviation of 1.2% over the 11 flights concerned). The most significant variability was found on the offset (coordinate at origine of the regression line), which varied between -4.6 and -1.9 °C , with a standard deviation of 2.6 °C. This variation may be explained by the fine wire resistance varying with time due to oxydation. From this calibration, and due to the incertitude of the housing features and recovery factor, we applied the same recovery factor of the Rosemount (0.98), to retrieve the static temperature from the impact temperature. Those results were similar to those found in the analysis of Baehr et al. 2002 on the same type of fine wire, and same antenna.

We added those details about the sensor in section 3:
*"During EUREC4A, temperature was also measured using two fine wires (Baehr et al. 2002) that were housed in a tubular antenna. The two platinum fine wires are housed in a tubular antenna from SFIM company (model T4113). They are more directly exposed to the stream, but protected from radiation, which consequently should not have a significant impact."*

A clarification about the Rosemount probe behaviour has been added:
*"Those were not easily explained, but supposed to be inherent to the sensor itself. Rarely, a large noise could also appear locally in the presence of cloud droplets."*

And also more discussion in section 5 about the fine wire:
*"The two fine wires were installed starting at flight RF09, and calibrated with the Rosemount probe at 1Hz for each flight. Both fine wires were consistent together, but one showed some noise that the other did not show at all. We consider only the latter here. We considered this measurement as non-absolute, and used it only for the study of temperature fluctuations. We calibrated the fine wire with the raw impact temperature of the Rosemount probe temperature as a reference, with one calibration per flight. The regression slope was very close to 1 (1.07 in average, with a standard deviation of 1.2\% over the 11 flights concerned). The most significant variability was found on the offset (coordinate at origin of the regression line), which varied between -4.6 and -1.9 $^\circ$C , with a standard deviation of 2.6 $^\circ$C. This variation may be explained by the fine wire resistance varying with time due to oxidation. From this calibration, and due to the incertitude of*

*the housing features and recovery factor, we applied the same recovery factor of the Rosemount (0.98), to retrieve the static temperature from the impact temperature. Those results were similar to those found in the analysis of Baehr et al. (2002) on the same type of fine wire, and same antenna."*

references:
- Baehr C., Méquignon A., Piguet B., 2002: Une première approche du capteur de température à fils fins sur le Merlin IV. Rapport interne, Météo-France/CNRM/GMEI/TRAMM.

**Technical/Minor Comments-**
1. Line 7 (abstract) delete 'a fast rate' and simply replace with '25 Hz'
The correction has been made.

2. Line 45 'Section 2 describes…'
The correction has been made.

3. Figure 1 – in caption, note that R-pattern is shown in red and L-pattern is shown in blue. Remove reference to S-Pattern. The last sentence in the figure description does not make sense to me...I'm not sure what the authors are trying to convey.
Thank you for your recommendations. The last sentence regarding the surface leg has been removed for simplification and clarification.

4. Table 1 – The authors should provide some description about the shorthand being used—it took me a while to figure out that, for example, R strati (1830 m) + 2R cb (680m – 740 m) referred to "1 R pattern in stratiform clouds at 1830 m and 2 R patterns at cloud base, one at 680 m and the other at 740 m" --- I still don't know what L flower is?
We agree with Reviewer 1 that Table 1 was not sufficiently explicit. To avoid confusion, the notations 'strati', 'top' and 'flower' have been merged into a single notation, 'strati', because in all of those cases, the leg was performed in the anvil of the cloud. In order to clarify this point, the following sentence has been added in the caption of Table 1:
*"The flight altitude is indicated between brackets and the notation 'cb', 'strati' and 'surf' refer to cloud base, stratiform layer and surface, respectively."*

5. Line 87-88 – Inertial navigation unit (Xsea model)? Is this a manufacturer and model number? I've never heard of that and didn't find anything with a quick internet search.
The INS system is AIRINS (model 6005214) from Ixblue company. The followig clarification has been made:
*"The ground velocity is measured with inertial navigation unit (AIRINS, model 6005214 from Ixblue company)."*

6. Line 126 – Krypton
The correction has been made.

7. Line 142 – KH20 showed 'a very good behavior.' -- What do you mean by this? It tracked well with other measures? Authors need to be more descriptive here.
We understand that the expression 'a very good behavior' can seem vague here, especially prior to the analysis which follows and more precisely explains what we consider as a "good behaviour". The performances of the KH20 were progressively improved during the campaign thanks to the

feedbacks and evaluations done after each flight. Therefore, the paragraph in Section 3 'Aircraft in situ instrumentation' about the KH20 issues, has been modified to clarify this point:

*"The KH20 also showed issues during this first phase, partly due to the particular conditions of the marine environment encountered during EUREC4A, which make it very challenging to measure air moisture at fine scale. The drastic change of water vapour content from above the inversion (where relative humidity can be as dry as a few percent) to below cloud base (where relative humidity is generally higher than 80%), was a challenge and the spacing between the emitter and the receiver of the KH20 sensor has been adjusted. In the subcloud layer patterns, the sea salt loading of the KH20 sensor generated a significant loss of signal dynamics. An assiduous cleaning of the optics at the beginning of each flight allowed to limit this loss of signal. Regarding the KH20 behaviour, many technical issues have been gradually solved and several improvements have been made following the feedbacks at the end of each flight. Thus, the KH20 performances have been significantly improved by the second phase of the campaign (flights RF09 to RF19). The calibration of moisture fluctuations, choice of reference slow measurement and the relative performances of the KH20 and Licor are discussed further in Section 4."*

Also, at the beginning of Section 4, the expression "during which the KH20 showed a very good behaviour" has been replaced by:

*"[…] during which the KH20 showed a very good behaviour, in terms of time response, and of consistency with other moisture sensors".*

8. Line 268 – orthogonal? (not orthonormed...)
The correction has been made.

---

## Author Comment (AC2)

**Answer to reviewer # 2**
"The EUREC⁴A turbulence dataset derived from the SAFIRE ATR 42 aircraft"
Pierre-Etienne Brilouet, Marie Lothon, Jean-Claude Etienne, Pascal Richard, Sandrine Bony, Julien Lernoult, Hubert Bellec, Gilles Vergez, Thierry Perrin, Julien Delanoë, Tetyana Jiang, Frédéric Pouvesle, Claude Lainardn, Michel Cluzeau, Laurent Guiraud, Patrice Medina, Theotime Charoy

**The authors would like to thank the anonymous reviewer for his/her suggestions and relevant remarks, which helped us to improve the manuscript. The original text from the review is written in black below, our reply in blue and the proposed modifications of the manuscript in red.**
* * *
This is a very interesting and useful dataset addressing the issue of cloud cover in the trade-wind region and its consequent variation in albedo -- among other things. The temperature and humidity content of the air was ably measured by sophisticated sensors of both fast response and slow (stable) response. Considerable care was taken in testing and calibrating the instruments to ensure good absolute accuracy over the frequency range from 0 Hz to 12.5 Hz (25 samples per second). The wind determination over the similar bandwidth was not discussed, resumably because it is already well characterized by SAFIRE.

This is a relevant remark. We did not detail the velocity field determination because the method has been verified in numerous field campaigns and the SAFIRE research team has a robust expertise on this measure. In order to consolidate and clarify this aspect, the following sentence has been added in section 3 :

*"The velocity measurement and computation has proved reliable in numerous field campaigns (Lambert and Durand , 1998 ; Saïd et al., 2005, Saïd et al. 2010)".*

In Figure 2b some variance appears to be forgone in the method of calibration of the (fast) KH20 using the (0.4 Hz) WVSS2 and the (1Hz) 1011C as (competing) references. The cyan trace (1011C) is visible both above the pink trace on crests and below the pink trace in troughs. That may not be noise. Likewise in the spectra of Figure 6 the KH20 and the Li-Cor separate at about 0.4 Hz, just about the report frequency of the WVSS2. I would trust the Li-Cor at least up to 2 Hz. I recommend checking out complementary filtering as a way to link the WVSS2 at low frequencies to the KH20 at higher frequencies. Assuming the WVSS2 data are available from all of the EUREC 4 A flights, this approach is possible using the existing data.

A critical issue in the study was to determine which slow sensor should be most suitable as a reference for calibrating the fast sensors. As you noticed, in Fig. 2b, the amplitudes of the signal calibrated with the WVSS2  is smaller than with the 1011C, suggesting a loss of variance. This, as you also suggested, is not noise. Figure R1 presents the variance of water vapor mixing ratio computed with the KH20 signal calibrated with the 1011C sensor versus the KH20 signal calibrated with the WVSS2 sensor. The variances computed with the 1011C calibration are indeed higher. Nevertheless, after analysis of several legs, we hypothesized that this is actually an overestimation of the variance, when the fast sensor was calibrated by the 1011C sensor.

[Figure]

Figure R1 : Variance of water vapor mixing ratio computed with the KH20 signal calibrated with the 1011C sensor versus the KH20 signal calibrated with the WVSS2 sensor. The comparison is done for all flight legs from RF09 to RF19.

It appeared that the sensor was very affected by the particular sampling conditions. The principle of the measurement is to determine the dew point with a chilled-mirror but this approach is not easily compatible with sudden changes in humidity. It can also be contaminated by liquid water when passing through cloud. More details will be available in Etienne et al. 2021, about the Core in situ data measurement of the ATR 42, currently in preparation.

Therefore, a figure has been added (Fig. 3) showing an example of a leg on which the 1011C does not handle abrupt moisture transitions well, resulting in a signal amplitude that is greatly overestimated and does not characterize a physical process. The discussion about this should thus been clearer than in the previous version.

[Figure]

*Figure 3 : Same as Fig. 2 or a leg (l1c) flown at z$\sim$600 m on 13 February 2020 (RF19).*

The paragraph regarding the justification of the slow sensor choice has been thus modified to better highlight the deficiencies of the 1011C :

*«The latter, despite its smaller response time, showed more difficulties in following the large variability of air moisture encountered during EUREC4 A, which added to the challenges of measuring air moisture in an environment with sea salt, clouds or even rain. This phenomenon can be noticed around 10:29:20 UTC in Fig. 2b and more clearly in Fig. 3, where the 1011C signal (dashed blue) shows several exagerated peaks, because it responded too slowly to the increasing and following fast levelling of moisture. This behaviour is explained by its measurement principle, with condensation at the mirror surface, which requires time to recover by drying. This issue resulted in a positive bias of about 27 % in the estimated moisture variance when the KH20 was calibrated with the 1011C hygrometer. This bias is visible in Fig. 2b and even more clearly in Fig. 3b from the difference of fluctuation energy between the two signals.»*

We believe that the reviewer's remark about the spectra from Licor and KH20 signals is independent from the first calibration issue adressed before. To support this, we have checked whether the difference in energy density spectrum, bewteen the Licor and the KH20 sensors, was impacted by the choice of the slow reference. As shown in Figure R2, the behaviour of the fast sensor in the inertial domain is independent of the choice of the slow sensor: for both calibrations, the spectra of KH20 and Licor agree together at smaller frequencies, and depart at 0.4 Hz. The only difference that is seen from one calibration to the other is a shift of energy, or variance bias discussed above.

[Figure]

Figure R2 : Comparison of Licor and KH20 spectra calibrated either from the 1011C sensor and the WVSS2 sensor.

As a conclusion, it is possible that we underestimate the variance when using the WVSS2 as a reference instead of the 1011C. But due to the problems on the 1011C, which obviously implied an overestimation of the variance on several legs, we prefered not to correct the variance obtained with

the WVSS2 calibration. In any case, this would not change the frequency where Licor and KH20 energy spectra depart from each other.

The data set is fully acceptable as it is, but the opportunity to pick up some additional variance, and hopefully covariance, may be attractive.

The color-coded flag system of figure 5 and Table 4 are very helpful as is the organization into characterized and defined ("stabilized") flight segments 30 km, 60 km, and longest possible (ragged sizes longer than 60 km). Turbulent departures are provided in two modes: detrended over a whole segment or high-pass filtered to pass only departures shorter than about 5 km (the ogive length). This two-tier system looks like a good way to supply turbulent departures for use by other researchers, especially for the strongly heterogeneous segments gathered from cloud base.

Figure 1: Useful to identify "R" as the red pattern and "L" as the blue pattern.
Thank you for your recommendation, which is shared by the other reviewer. The following modification in the figure caption has been made:
*"R-pattern is shown in red and L-pattern is shown in blue".*

Table 1: Several edits: ShCu, StCu should be expanded in caption. Explain or define "flower clouds", L surf , L flower , L top , R cb , maybe others
In order to clarify the abbreviations related to the cloud cover, the following note has been added at the bottom of Table 1:
*"For the description of the cloud cover, the abbreviations are defined as follow Cu : Cumulus, ShCu : Shallow Cumulus, StCu : Stratocumulus and Flower clouds : Circular clumped patterns as introduced by Stevens et al. (2020)".*
Also, the Table caption has been completed to properly define the abbreviations in the flight strategy column:
*"The flight altitude is indicated between brackets and the "cb", "strati" and "surf"notations refer to "cloud base", "stratiform layer" and "surface", respectively."*

*Line 85: better to call 4 m the "sample spacing." The word "resolution" is somewhat ambiguous.*
The correction has been made.

Line 89: The angles of attack and sideslip are not the Euler angles. The Euler angles (roll, pitch, and yaw) describe the orientation of the aircraft with respect to the earth. The angles of attack and sideslip describe the orientation of the multiport (nose-cone) probe to the oncoming airstream in flight. This appears to be an editing issue rather than a sign of error in the actual calculations. It can be addressed most simply by consulting a team member who has made such calculations.
Thank you for pointing out this mistake. The incorrect mention of the Euler angles has been removed:
*"The velocity of the air relative to the aircraft is computed from the measurement of the true air speed magnitude, the attack and side slip angles, according to Lenschow (1986)."*

Line 152: subcloud (typographic error)
The correction has been made.

Line 202: Did you mean "lose" instead of "loose"?
Yes, the typing error has been corrected.

Figure 13 Needs editing to make the caption fit with the figure.
We apologize for the wrong figure caption. The following correction has been made:

*"Normalized vertical profiles of variance of (a) vertical velocity, (b) horizontal turbulent kinetic energy, (c) temperature and (d) water vapour mixing ratio. Flight numbers are indicated in the top right box. For the water vapour mixing ratio, only the legs with a green or a yellow combined flag have been considered. The normalized altitude z_\* is defined by z/LCL, where LCL is the lifting condensation level."*

Figure 14 Same: Also include definition of Z\* in at least one of these figures

As for your previous remark, we apologize for this missing information The definition of the normalized altitude z\* has been added to the captions of the two figures. Also, the caption of Figure 14 has been updated as follow:

*"Normalized vertical profiles of (a) the heat flux, (b) the moisture flux, systematic error (c) for the heat flux, (d) for the moisture flux, random error (e) for the heat flux and (f) for the moisture flux. Flight numbers are indicated in the top right box. The normalized altitude z_\* is defined by z/LCL, where LCL is the lifting condensation level."*

Also, a clarification about the normalization of the profiles has been added :

*« The profiles are normalized by the lifting condensation level (LCL), estimated here as the flight altitude of the rectangle at the cloud base minus 50 m.»*

Bibliography: F. Saïd, G. Canut, P. Durand, F. Lohou, M. Lothon were not listed as authors of the reference given on line 423.

The correction has been made.